# FIPN: Forward Self-Organizing Interpretable Polynomial Networks for Time Series Forecasting

Yizhen Wang [1]   Zheng Wang [1 2]   Eun-Hu Kim [1]   Zunwei Fu [2]

## Abstract

Most existing time-series forecasters are trained end-to-end with backpropagation, which often increases computational cost and limits structural transparency. As a result, it remains difficult to trace how their predictions are formed. This paper presents FIPN, a forward self-organizing interpretable polynomial network for time-series forecasting. FIPN grows its architecture layer by layer through candidate generation, closed-form estimation, and validation-based selection, without relying on backpropagation-based optimization. Each neuron couples a fuzzy rule antecedent with a Fourier-enhanced polynomial consequent: fuzzy clustering softly partitions the input space and produces interpretable rule weights for local regimes, while the consequent retains access to original features and uses Fourier functions to capture periodic and frequency-related structures. Since forward growth may introduce redundancy, collinearity, and overfitting as depth increases, FIPN further incorporates regularized node scoring, dropout-based redundancy control, and persistent access to raw inputs. Experiments on long-horizon forecasting benchmarks show that FIPN achieves competitive accuracy with a compact model size, while the learned fuzzy rules provide rule- and structure-level explanations. These results suggest that forward self-organizing polynomial networks offer a practical balance among accuracy, efficiency, and interpretability for long-horizon forecasting.

## 1. Introduction

Time series forecasting is central to many real-world applications, including energy management, environmental monitoring, traffic control, and financial analysis. As data scale and complexity continue to increase, time series often exhibit multiple characteristics at the same time, including multi-scale periodicity, long-range dependence, and state changes over time. This growing complexity has driven rapid progress in deep sequence models. Recent deep sequence models, including recurrent networks, temporal convolutional networks, and Transformer-based architectures, have achieved strong performance on standard forecasting benchmarks (Lai et al., 2018; Vaswani, 2017; Zhou et al., 2021; Wu et al., 2021). However, most of these models are trained end-to-end by backpropagation, which often introduces substantial computational and memory costs. Moreover, their internal representations are highly distributed, making it difficult to trace how predictions are formed. These limitations are especially important in resource-constrained or high-stakes scenarios where both efficiency and structural transparency are required.

Polynomial Neural Networks (PNNs), originating from the Group Method of Data Handling (GMDH), offer a classical forward self-organizing alternative to backpropagation-trained deep models (Ivakhnenko, 1971). Unlike fixed architectures optimized end-to-end, GMDH-style models construct network structures layer by layer through candidate generation, closed-form coefficient estimation, and validation-based selection. Each candidate is typically a low-order polynomial neuron formed from input feature pairs, whose parameters can be efficiently estimated by least squares. This training paradigm avoids backpropagation and provides a transparent structure, making it attractive for industrial modeling, control systems, and small-sample learning. Recent deep PNNs further extend this idea to richer polynomial architectures and have shown competitive performance with substantially fewer parameters than conventional deep neural networks (Chrysos et al., 2022). These results highlight thepotential of forward self-organizing polynomial structuresas efficient and interpretable alternatives to deep models.

Despite these advantages, applying PNNs directly to mod-

[1]Research Center for Big Data and Artificial Intelligence, Linyi University, Linyi 276005, China [2]College of Information Communication Technology, University of Suwon, Hwaseong-si, Gyeonggi-do 18232, South Korea. Correspondence to: Zheng Wang <wangzheng@suwon.ac.kr>.

*Proceedings of the 43rd International Conference on Machine Learning*, Seoul, South Korea. PMLR 306, 2026. Copyright 2026 by the author(s).

ern time series forecasting remains challenging. As network depth increases, the number of candidate neurons grows rapidly, and many neurons tend to learn highly similar polynomial functions. This behavior introduces strong multicollinearity among neuron outputs and leads to substantial structural redundancy, which weakens generalization as the network deepens (Zhou et al., 2024). At the same time, traditional PNNs mainly operate in the time domain and lack inductive bias for common time series properties such as periodicity and multi-frequency oscillations. This is a critical limitation because many real-world time series, such as electricity load, weather, and traffic flow, contain strong seasonal or oscillatory structures. Recent studies indicate that incorporating frequency information can significantly improve long-term forecasting, and that Fourier bases or spectral representations are able to capture long-range structure with relatively few parameters. (Yi et al., 2023; Fan et al., 2023) However, most frequency-aware forecasting models still rely on backpropagation and rarely combine spectral modeling with forward self-organizing polynomial networks.

To address these issues, we propose FIPN, a Forward Self-Organizing Interpretable Polynomial Network for long-horizon time series forecasting. FIPN follows the forward structure learning paradigm of GMDH and deep PNNs while unifying frequency-aware modeling and rule-level interpretation within a single framework. Each FIPN neuron consists of a fuzzy rule antecedent and a Fourier-enhanced polynomial consequent. The fuzzy antecedent softly partitions the input space and describes local states with clear rule semantics. The Fourier-enhanced consequent operates directly on raw input features, allowing low-order polynomials to effectively represent periodic and oscillatory behavior that is more naturally expressed in the frequency domain (Tancik et al., 2020). At the structural level, the model applies regularized node scoring, node-level dropout, and controlled forward growth to regulate model complexity. These mechanisms explicitly suppress redundancy and mitigate multicollinearity as the network deepens.

Extensive experiments on eight long-horizon forecasting benchmarks show that FIPN achieves competitive accuracy with a compact model size and low computational cost. Ablation studies further verify the contributions of Fourier-enhanced consequents, fuzzy rule gating, and redundancy-controlled forward selection. Overall, FIPN provides a forward-grown, closed-form, and structurally interpretable alternative to backpropagation-trained forecasters, offering a practical balance among accuracy, efficiency, and interpretability.

The main contributions of this work are summarized as follows:

- **A forward self-organizing forecasting framework**

**without backpropagation-based optimization.** FIPN introduces a layer-wise learning paradigm based on candidate generation, closed-form parameter estimation, and validation-based selection. By embedding frequency inductive bias directly into structural growth, the framework improves the modeling of long-term periodic patterns while maintaining high training efficiency.

- **A fuzzy rule-gated Fourier-enhanced polynomial neuron.** The fuzzy rules describe local temporal states, while the Fourier-enhanced polynomial part captures periodic and oscillatory patterns. In this way, each neuron keeps an explicit form and is easier to inspect than a fully black-box unit.

- **A redundancy-controlled mechanism for stable forward growth.** FIPN combines persistent raw-input access, regularized node scoring, and node-level dropout to reduce candidate redundancy and multicollinearity, leading to more stable closed-form estimation and improved generalization.

## 2. Related Work

Deep sequence models dominate long-horizon time series forecasting. Recent work improves efficiency by modifying attention mechanisms or by injecting structural priors such as trend, seasonality, or frequency components. These designs demonstrate that frequency-aware representations and lightweight architectures can substantially improve long-term prediction performance (Vaswani, 2017; Zhou et al., 2021; Wu et al., 2021; Yi et al., 2024; Wang et al., 2024a). However, most of these models are still trained end-to-end with backpropagation. Their internal reasoning is often opaque, and interpretability is typically addressed through post-hoc analysis rather than being intrinsically embedded into the model structure.

Forward self-organizing polynomial networks provide a classical alternative to backpropagation-trained deep models. Originating from the Group Method of Data Handling (GMDH), these models grow layer by layer through candidate generation, coefficient estimation, and validation-based selection (Ivakhnenko, 1971). Each candidate is typically a low-order polynomial neuron, whose parameters can be efficiently estimated by least-squares fitting. Modern Polynomial Neural Networks (PNNs) extend this paradigm with deeper structures, richer node parameterizations, and more flexible feature compositions, improving representation power while retaining polynomial computational units (Chrysos et al., 2022). Recent polynomial models further show that high-order interactions can be compactly represented through hierarchical or factorized mappings, such as deep polynomial networks, and Taylor-map polynomial

neural networks (Ivanov & Ailuro, 2024). Nevertheless, these methods mainly focus on representation capacity and high-order interaction modeling. Frequency-aware temporal modeling, rule-level interpretability, and redundancy control have not been systematically unified within a forward-grown polynomial forecasting architecture.

Frequency-domain representations provide a compact way to describe periodic and long-range temporal structures in time series (Tancik et al., 2020). Meanwhile, fuzzy rule-based models offer interpretable descriptions of local regimes and decision logic. Recent fuzzy neural models further improve rule-based learning through fuzzy clustering, uncertainty-aware reasoning, and feature selection mechanisms (Takagi & Sugeno, 1985; Bezdek, 1980; Zhu et al., 2026; Wang et al., 2024b; Kim et al., 2024). However, frequency modeling and fuzzy rule interpretation are often developed as separate components, and many related models still rely on backpropagation-based optimization. Few studies integrate spectral representations, fuzzy rule semantics, and forward self-organizing polynomial growth into a single forecasting framework. FIPN addresses this gap by unifying forward-grown polynomial construction, Fourier-enhanced polynomial neurons, intrinsic fuzzy rule gating, and redundancy-controlled node selection. This design is tailored to long-horizon forecasting and explicitly balances accuracy, computational efficiency, and structural interpretability.

## 3. Methodology

In this section, we present the proposed FIPN framework, as illustrated in Figure 1. We first formalize the long-horizon forecasting task and notation, then introduce the fuzzy rule-gated Fourier neuron, the forward polynomial construction, and the regularized node selection mechanism with stochastic dropout.

**Problem Formulation and Notation.** We consider multivariate long-horizon time-series forecasting. Let $\mathbf{X} \in \mathbb{R}^{R \times T}$ denote a multivariate sequence with $R$ channels and $T$ time steps. Given a look-back window of length $L$, the goal is to predict the next $H$ time steps. Under the sliding-window protocol, at time index $t$, the input and target are defined as $\mathbf{X}_t = \mathbf{X}_{:, t-L+1:t} \in \mathbb{R}^{R \times L}$, $\mathbf{Y}_t = \mathbf{X}_{:, t+1:t+H} \in \mathbb{R}^{R \times H}$. We learn a forecasting function $f_{\boldsymbol{\theta}}$ by minimizing the prediction loss $\mathcal{L}(f_{\boldsymbol{\theta}}(\mathbf{X}_t), \mathbf{Y}_t)$. We denote by $\mathbf{x}_t \in \mathbb{R}^d$ the feature representation extracted from $\mathbf{X}_t$, and by $l$ the layer index of the forward-grown network.

**Overall Framework.** FIPN is a forward-grown polynomial network that expands its architecture layer by layer. At each layer, a pool of candidate neurons is generated from the current representation. Each candidate is fitted independently

by closed-form regularized regression and evaluated on validation data; only a compact set of top-ranked candidates is retained to form the next-layer representation. Each neuron combines a fuzzy-clustering antecedent, which provides normalized rule activations, with a Fourier-enhanced polynomial consequent, which uses low-order harmonic cosine bases with variance-scaled frequencies. This design enables regime-aware modeling of local dynamics and compact representation of periodic temporal structures. To stabilize forward growth, FIPN uses regularized node scoring and column dropout to suppress redundant and highly correlated candidates. This yields efficient training and inference without backpropagation-based optimization, while retaining rule- and spectral-level interpretability.

### 3.1. Fuzzy Rule-Gated Fourier Neuron

We define the basic computational unit of FIPN as a fuzzy rule-gated Fourier neuron, which combines soft fuzzy partitioning with local Fourier-enhanced regression in a single interpretable module. Let $\mathbf{x} \in \mathbb{R}^d$ denote the feature vector extracted from a multivariate time-series window. For sample $n \in \{1, \ldots, N\}$, we denote the input and target as $\mathbf{x}_n \in \mathbb{R}^d$ and $\mathbf{y}_n \in \mathbb{R}^p$, respectively, where $p$ is the output dimension. Stacking all samples gives $\mathbf{X} = [\mathbf{x}_1^\top; \ldots; \mathbf{x}_N^\top] \in \mathbb{R}^{N \times d}$ and $\mathbf{Y} = [\mathbf{y}_1^\top; \ldots; \mathbf{y}_N^\top] \in \mathbb{R}^{N \times p}$.

We first obtain a fuzzy $C$-partition of the input space using fuzzy $c$-means (FCM), where $C$ denotes the number of fuzzy rules. FCM estimates a membership matrix $\mathbf{U} = [u_{nq}] \in [0, 1]^{N \times C}$ and a set of prototypes $\mathbf{V} = \{\mathbf{v}_q\}_{q=1}^{C}$, with $\mathbf{v}_q \in \mathbb{R}^d$. Given a fuzzification coefficient $\gamma > 1$, FCM solves

$$\min_{\mathbf{U}, \mathbf{V}} \sum_{n=1}^{N} \sum_{q=1}^{C} u_{nq}^{\gamma} \|\mathbf{x}_n - \mathbf{v}_q\|_2^2, \quad \text{s.t.} \sum_{q=1}^{C} u_{nq} = 1 \quad (1)$$

To connect fuzzy partitions with Fourier-enhanced consequents, we define the normalized rule gate as

$$\beta_{nq} = \frac{u_{nq}^{\gamma}}{\sum_{r=1}^{C} u_{nr}^{\gamma}}, \qquad \sum_{q=1}^{C} \beta_{nq} = 1, \quad (2)$$

where $\beta_{nq}$ measures the activation strength of rule $q$ for sample $n$.

We then construct rule-specific harmonic cosine bases centered at the fuzzy prototypes. For rule $q \in \{1, \ldots, C\}$ and input dimension $i \in \{1, \ldots, d\}$, let $v_{q,i}$ be the $i$-th component of $\mathbf{v}_q$. The local scale of dimension $i$ under rule $q$ is estimated by

$$s_{q,i}^2 = \frac{\sum_{n=1}^{N} u_{nq}^{\gamma} (x_{n,i} - v_{q,i})^2}{\sum_{n=1}^{N} u_{nq}^{\gamma}} + \varepsilon, \quad (3)$$

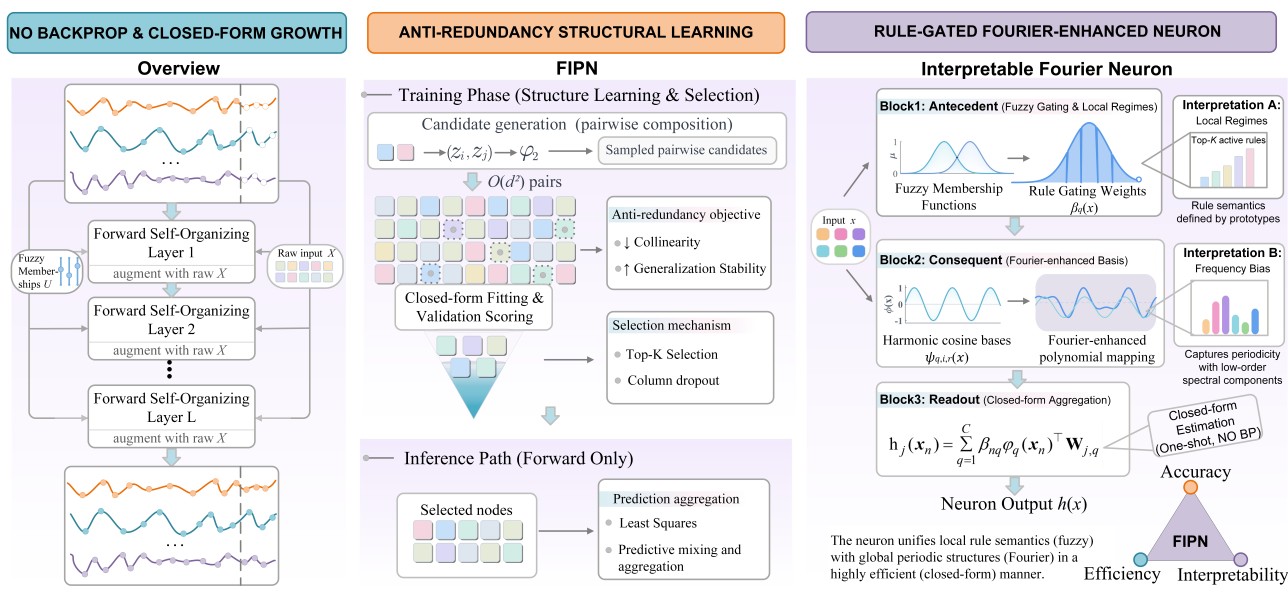

*Figure 1.* Overall architecture of FIPN. The model grows layer by layer through candidate generation, closed-form ridge fitting, and regularized validation selection, without backpropagation-based optimization. Each neuron combines fuzzy rule gates for local regime description with Fourier-enhanced polynomial consequents for periodic temporal modeling. The final selected representation supports compact, efficient, and interpretable long-horizon forecasting.

where $\varepsilon > 0$ is a small constant for numerical stability. The corresponding angular frequency is defined as

$$\omega_{q,i,r} = \frac{r}{s_{q,i}}, \qquad r \in \{1, \ldots, r_{\max}\}. \quad (4)$$

The local harmonic basis is then given by

$$\psi_{q,i,r}(\mathbf{x}_n) = \cos\left(\omega_{q,i,r}\left(x_{n,i} - v_{q,i}\right) + \phi_{q,i,r}\right), \quad (5)$$

where $\phi_{q,i,r} \in [0, 2\pi)$ denotes the phase offset. The inverse-variance scaling $\omega_{q,i,r}$ enhances frequency resolution for locally stable dimensions while suppressing noisy ones, yielding a rule-adaptive and band-limited spectral representation. Using the rule gates and Fourier-polynomial features, the output of candidate neuron $j$ is

$$\mathbf{h}_j(\mathbf{x}_n) = \sum_{q=1}^{C} \beta_{nq} \boldsymbol{\varphi}_q(\mathbf{x}_n)^\top \mathbf{W}_{j,q}, \quad (6)$$

where $\boldsymbol{\varphi}_q(\mathbf{x}_n)$ is the local Fourier-polynomial feature vector and $\mathbf{W}_{j,q}$ is the rule-specific consequent coefficient matrix. This formulation makes each neuron a fuzzy rule-weighted mixture of local Fourier-polynomial experts.

### 3.2. Forward Polynomial Network Construction

FIPN constructs a polynomial network through layer-wise forward growth and node selection. At each layer, candidate neurons are generated from the current representation, fitted independently, evaluated on validation data, and selectively

---

**Algorithm 1** Training FIPN

1: **Input:** $(\mathbf{X}^{tr}, \mathbf{Y}^{tr}), (\mathbf{X}^{va}, \mathbf{Y}^{va}), L^\star, \{K_l\}, C, r, \lambda, \rho, \alpha$
2: **Output:** $\{\mathcal{S}_l\}_{l=1}^{L^\star}$ and $\mathbf{H}^{(L^\star)}$
3: **Init:** $\mathbf{H}^{(0)} \leftarrow \mathbf{X}$
4: **for** $l = 0$ to $L^\star - 1$ **do**
5: $\quad \widetilde{\mathbf{H}}^{(l)} \leftarrow [\mathbf{X}, \mathbf{H}^{(l)}]; \quad \mathcal{C}_{l+1} \leftarrow \text{GENCAND}(\widetilde{\mathbf{H}}^{(l)})$
6: $\quad$ **for all** $j \in \mathcal{C}_{l+1}$ **do**
7: $\quad\quad (\mathbf{F}_j^{tr}, \mathbf{F}_j^{va}) \leftarrow \text{BUILDDESIGN}(j, C, r_{\max})$
8: $\quad\quad \widetilde{\mathbf{F}}_j^{tr} \leftarrow \text{COLDROP}(\mathbf{F}_j^{tr}, \rho)$
9: $\quad\quad \mathbf{W}_j^\star \leftarrow \text{RIDGECF}(\widetilde{\mathbf{F}}_j^{tr}, \mathbf{Y}^{tr}, \lambda)$
10: $\quad\quad \mathcal{J}_j \leftarrow \text{SCORE}(\mathbf{F}_j^{va}, \mathbf{Y}^{va}, \mathbf{W}_j^\star, \alpha)$
11: $\quad$ **end for**
12: $\quad \mathcal{S}_{l+1} \leftarrow \text{TOPK}(\{\mathcal{J}_j\}_{j \in \mathcal{C}_{l+1}}, K_{l+1})$
13: $\quad \mathbf{H}^{(l+1)} \leftarrow [\mathbf{h}_j^{(l+1)}]_{j \in \mathcal{S}_{l+1}}$
14: **end for**

---

retained to form the next-layer input. This process incrementally increases network depth while maintaining a strictly feedforward structure.

Let the input representation to layer $l$ be denoted as

$$\mathbf{H}^{(l)}(\mathbf{X}) = \left[\mathbf{h}_1^{(l)}(\mathbf{X}), \ldots, \mathbf{h}_{M_l}^{(l)}(\mathbf{X})\right] \in \mathbb{R}^{N \times M_l}, \quad (7)$$

where $M_l$ is the number of retained neurons at layer $l$, and $\mathbf{h}_j^{(l)}(\mathbf{X}) \in \mathbb{R}^{N \times 1}$ is the output vector of neuron $j$ evaluated on all $N$ samples. At the initial layer, $\mathbf{H}^{(0)}$ is given by the original feature matrix. To preserve direct access to raw inputs, each layer can generate candidate neurons from both

the original features and the retained representation of the previous layer.

Given the design matrix $\mathbf{F}_j$ for candidate neuron $j$, we estimate its multi-output regression coefficients $\mathbf{W}_j \in \mathbb{R}^{D_j \times p}$ on the training set $(\mathbf{F}_j^{\mathrm{tr}}, \mathbf{Y}^{\mathrm{tr}})$ via ridge-regularized least squares:

$$\mathbf{W}_j^\star = \arg \min_{\mathbf{W}} \left\| \mathbf{F}_j^{\mathrm{tr}} \mathbf{W} - \mathbf{Y}^{\mathrm{tr}} \right\|_F^2 + \lambda \left\| \mathbf{W} \right\|_F^2, \quad (8)$$

which admits the closed-form solution

$$\mathbf{W}_j^\star = \left( (\mathbf{F}_j^{\mathrm{tr}})^\top \mathbf{F}_j^{\mathrm{tr}} + \lambda \mathbf{I} \right)^{-1} (\mathbf{F}_j^{\mathrm{tr}})^\top \mathbf{Y}^{\mathrm{tr}}. \quad (9)$$

After all candidates at layer $l$ are fitted, FIPN ranks them according to the validation score introduced in the next subsection. Only the top-ranked candidates are retained to construct $\mathbf{H}^{(l+1)}$. This generate-fit-validate-select process is repeated until the final depth $L^\star$ is reached. The final representation $\mathbf{H}^{(L^\star)}$ is then used for forecasting. Therefore, FIPN does not aggregate predictions from all intermediate layers; instead, it progressively selects useful nodes and composes them into the final forward-grown representation.

### 3.3. Regularized Node Selection with Dropout

As the network grows deeper, forward composition may produce candidate neurons with highly similar outputs. This increases redundancy and multicollinearity, and may make closed-form estimation unstable. To alleviate this problem, FIPN combines column dropout with regularized node scoring, so that each layer retains only a compact set of accurate and diverse candidates. The overall training and selection procedure is summarized in Algorithm 1.

For candidate neuron $j$, let $\mathbf{F}_j^{\mathrm{tr}} \in \mathbb{R}^{N_{\mathrm{tr}} \times D_j}$ be its training design matrix, where $D_j$ is the number of consequent features. We sample a feature-wise binary mask $\mathbf{d}_j \in \{0, 1\}^{D_j}$ and share it across all training samples. The column dropout operation is defined as

$$d_{j,m} \sim \mathrm{Bernoulli}(1 - \rho), \quad (10)$$

$$\widetilde{\mathbf{F}}_j^{\mathrm{tr}} = \mathbf{F}_j^{\mathrm{tr}} \odot (\mathbf{1} \mathbf{d}_j^\top), \quad (11)$$

Here, $\rho \in [0, 1)$ is the dropout rate, $\mathbf{1} \in \mathbb{R}^{N_{\mathrm{tr}} \times 1}$ is an all-ones vector, and $\odot$ denotes element-wise multiplication. Since the same mask is shared by all samples, dropout acts on feature columns rather than samples. The ridge solution is then computed using $\widetilde{\mathbf{F}}_j^{\mathrm{tr}}$, which reduces the tendency of a candidate to rely on a small group of correlated features.

Candidate selection is based on a regularized validation score:

$$\mathcal{J}_j = \frac{1}{N_{\mathrm{va}}} \left\| \mathbf{Y}^{\mathrm{va}} - \mathbf{F}_j^{\mathrm{va}} \mathbf{W}_j^\star \right\|_F^2 + \alpha \left\| \mathbf{W}_j^\star \right\|_F^2, \quad (12)$$

where $\alpha \geq 0$ controls the coefficient penalty. The first term measures validation error, while the second term discourages overly large coefficients, which are often associated with noise sensitivity and ill-conditioning.

At layer $l + 1$, the retained node set is selected as

$$\mathcal{S}_{l+1} = \mathrm{TopK}\left( \{\mathcal{J}_j\}_{j \in \mathcal{C}_{l+1}}, K_{l+1} \right), \quad (13)$$

where $\mathcal{C}_{l+1}$ is the candidate pool and $\mathrm{TopK}(\cdot)$ returns the indices of the $K_{l+1}$ lowest-scoring candidates. The next-layer representation is then

$$\mathbf{H}^{(l+1)}(\mathbf{X}) = \left[ \mathbf{h}_j^{(l+1)}(\mathbf{X}) \right]_{j \in \mathcal{S}_{l+1}}. \quad (14)$$

This generate-fit-validate-select procedure keeps the network strictly forward-grown while controlling width and complexity. By retaining only high-utility candidates at each layer, FIPN gradually forms the final representation $\mathbf{H}^{(L^\star)}$ in a stable and interpretable manner.

## 4. Experiments

**Dataset.** We evaluate FIPN on eight widely used multivariate long-horizon forecasting benchmarks covering energy, traffic, weather, and finance. The evaluation suite includes the ETT family, namely ETTh1, ETTh2, ETTm1, and ETTm2 (Zhou et al., 2021), as well as Electricity, Traffic, Weather (Wu et al., 2021), and Exchange Rate. Following common practice (Liu et al., 2023), we adopt chronological data splits with a 6:2:2 train/validation/test ratio for the ETT datasets and a 7:1:2 ratio for the remaining datasets. Detailed dataset statistics are provided in Appendix A.1.

**Baselines.** We compare FIPN with representative long-horizon forecasting models, including efficient architectures and Transformer-based methods. The compared methods include TimeStacker (Liu et al., 2025), a frequency-aware multilevel stacking framework, FilterTS (Wang et al., 2025), which performs comprehensive frequency filtering for multivariate forecasting, and SeqComp (Chen et al., 2025), which enhances representations with learnable complementary sequences. We also consider TimeMixer (Wang et al., 2024a), a decomposable multiscale MLP-based mixing model, and FITS (Xu et al., 2024), a lightweight frequency-domain model based on complex-valued interpolation. In addition, we evaluate representative Transformer variants, including iTransformer (Liu et al., 2023), PatchTST (Nie, 2022), and Crossformer (Zhang & Yan, 2023). DLinear (Zeng et al., 2023) is included as a standard linear baseline.

**Implementation Details.** All experiments are conducted under a unified pipeline on a workstation with an Intel(R) Core(TM) i9-12900K CPU (3.20 GHz) and an NVIDIA RTX4060 GPU (16GB), running Windows 10. FIPN is implemented in MATLAB R2020b. We fix the look-back

*Table 1.* Long-term forecasting results averaged over four prediction horizons {96, 192, 336, 720} with input length 96. The best results are in red and the second best are blue. Full horizon-wise results are reported in Appendix C.

| Models | FIPN (ours) | | TimeStacker (2025) | | FilterTS (2025) | | SeqComp (2025) | | TimeMixer (2024) | | FITS (2024) | | iTransformer (2024) | | PatchTST (2023) | | Crossformer (2023) | | DLinear (2023) | |
|---|---|---|---|---|---|---|---|---|---|---|---|---|---|---|---|---|---|---|---|---|
| Metric | MSE | MAE | MSE | MAE | MSE | MAE | MSE | MAE | MSE | MAE | MSE | MAE | MSE | MAE | MSE | MAE | MSE | MAE | MSE | MAE |
| ETTh1 | **0.428** | **0.422** | 0.433 | _0.423_ | 0.433 | 0.430 | _0.429_ | 0.435 | 0.449 | 0.442 | 0.451 | 0.440 | 0.454 | 0.447 | 0.469 | 0.455 | 0.529 | 0.522 | 0.456 | 0.452 |
| ETTh2 | **0.361** | _0.391_ | 0.368 | **0.390** | 0.372 | 0.396 | 0.368 | 0.398 | _0.364_ | 0.395 | 0.383 | 0.408 | 0.383 | 0.407 | 0.387 | 0.407 | 0.942 | 0.684 | 0.559 | 0.515 |
| ETTm1 | _0.385_ | 0.399 | **0.381** | **0.381** | _0.385_ | 0.396 | **0.381** | 0.398 | **0.381** | _0.395_ | 0.415 | 0.405 | 0.407 | 0.410 | 0.387 | 0.400 | 0.496 | 0.496 | 0.403 | 0.407 |
| ETTm2 | **0.267** | **0.313** | _0.274_ | _0.316_ | 0.276 | 0.321 | 0.275 | 0.323 | 0.275 | 0.323 | 0.286 | 0.408 | 0.288 | 0.332 | 0.281 | 0.326 | 0.757 | 0.610 | 0.350 | 0.401 |
| Traffic | _0.441_ | **0.276** | 0.508 | 0.335 | 0.471 | 0.315 | 0.472 | 0.301 | 0.484 | 0.297 | 0.627 | 0.376 | **0.428** | _0.282_ | 0.481 | 0.304 | 0.550 | 0.304 | 0.625 | 0.383 |
| Electricity | **0.168** | **0.263** | 0.194 | 0.275 | 0.180 | 0.271 | 0.192 | 0.282 | 0.182 | 0.272 | 0.217 | 0.295 | _0.178_ | _0.270_ | 0.205 | 0.290 | 0.244 | 0.334 | 0.212 | 0.300 |
| Weather | _0.242_ | _0.270_ | 0.243 | **0.264** | 0.244 | 0.274 | 0.243 | 0.273 | **0.240** | 0.271 | 0.249 | 0.276 | 0.258 | 0.278 | 0.259 | 0.348 | 0.259 | 0.315 | 0.265 | 0.317 |
| Exchange | 0.356 | 0.401 | **0.336** | **0.389** | _0.352_ | _0.397_ | 0.356 | 0.400 | 0.355 | 0.399 | 0.353 | 0.399 | 0.360 | 0.403 | 0.367 | 0.404 | 0.940 | 0.707 | 0.354 | 0.414 |

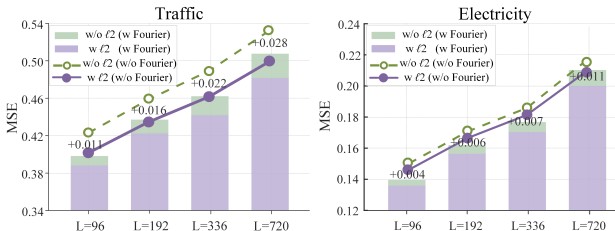

*Figure 2.* Horizon-wise effect of $\ell_2$-regularized closed-form estimation on Traffic and Electricity. Bars show FIPN with Fourier-enhanced consequents, with shaded regions indicating the $\ell_2$ gain; solid and dashed lines denote the non-Fourier variant with and without $\ell_2$ regularization.

length to $L = 96$ and evaluate four forecasting horizons, $H \in \{96, 192, 336, 720\}$. For FIPN, candidate neurons are fitted by ridge-regularized closed-form regression, and model selection is performed on the validation set. The maximum forward construction depth, retained width $\{K_l\}$, number of fuzzy rules $C$, maximum harmonic order $r_{\max}$, ridge parameter $\lambda$, dropout rate $\rho$, and scoring weight $\alpha$ are selected from a predefined search space. Unless otherwise specified, all reported results are obtained under the same data splits and evaluation protocol. The full implementation is included in the supplementary code package "FIPN CODE."

### 4.1. Experimental Results

Table 1 reports the main results on eight benchmarks with input length 96, where MSE and MAE are averaged over four prediction horizons. In the table, bold red and underlined blue denote the best and second-best results, respectively. FIPN achieves strong overall performance and obtains the largest number of first-place results among the compared methods. Its advantage is most evident on ETT, Electricity, and Weather, whose temporal patterns contain clear periodic and multi-scale structures. This is consistent with the model design: Fourier-enhanced consequents capture seasonal and oscillatory components, while fuzzy rule gates adapt local polynomial mappings to dif-

ferent temporal regimes. Compared with DLinear and recent efficient forecasters, FIPN reduces errors on most datasets, indicating that a compact forward-grown polynomial structure can capture nonlinear temporal dependencies without backpropagation-based optimization. It also remains competitive with strong Transformer-based models such as iTransformer and PatchTST, while retaining a smaller and more interpretable structure. On Traffic and Exchange, where the data are more event-driven or weakly predictable, the performance gaps among leading methods are smaller. FIPN still stays close to the best results, suggesting stable behavior beyond strongly periodic datasets. A depth-wise utility analysis in Appendix B.4 further shows that FIPN remains stable at greater depths, whereas classical PNN/GMDH-style baselines deteriorate earlier under greedy forward growth.

### 4.2. Ablation Study

**Ablation of the Fourier-Enhanced Consequents.** We first remove the harmonic cosine bases and retain only the polynomial consequent, denoted as w/o Fourier. As shown in Table 2, this variant causes the most pronounced degradation, increasing the average MSE and MAE by 5.50% and 4.30%, respectively. This result confirms that the Fourier-enhanced consequent is the most important component for long-horizon forecasting. Without the spectral basis, low-order polynomial terms must implicitly approximate periodic and oscillatory patterns, which is less efficient and becomes more difficult at longer horizons. The degradation is especially evident on datasets with clear temporal periodicity, such as ETT, Electricity, and Weather, supporting the role of Fourier bases in capturing multi-scale temporal dynamics.

**Ablation of Node Selection and Dropout.** Removing the redundancy-controlled selection/dropout mechanism also leads to consistent performance degradation, with average increases of 2.70% in MSE and 2.20% in MAE. This observation is consistent with the forward-grown polynomial construction of FIPN. As the network grows deeper, can-

*Table 2.* Component ablation study of FIPN. We evaluate the effect of removing Fourier-enhanced consequents, redundancy-controlled selection/dropout, and fuzzy rule gating. MSE and MAE are averaged over four prediction horizons on eight benchmarks and reported as mean$\pm\sigma$ over three random seeds. The last row gives the average error across datasets, and the percentage in parentheses denotes the relative improvement of the full model over each ablated variant.

| Dataset | w/o Fourier | | w/o Selection/Dropout | | w/o Rule Gating | | Full Model | |
|---|---|---|---|---|---|---|---|---|
| | MSE | MAE | MSE | MAE | MSE | MAE | MSE | MAE |
| ETTh1 | 0.4515$\pm$0.0035 | 0.4401$\pm$0.0026 | 0.4396$\pm$0.0031 | 0.4313$\pm$0.0023 | 0.4327$\pm$0.0026 | 0.4262$\pm$0.0019 | 0.4280$\pm$0.0017 | 0.4220$\pm$0.0015 |
| ETTh2 | 0.3809$\pm$0.0024 | 0.4078$\pm$0.0019 | 0.3707$\pm$0.0021 | 0.3996$\pm$0.0017 | 0.3650$\pm$0.0018 | 0.3949$\pm$0.0014 | 0.3610$\pm$0.0012 | 0.3910$\pm$0.0008 |
| ETTm1 | 0.4062$\pm$0.0028 | 0.4162$\pm$0.0022 | 0.3954$\pm$0.0025 | 0.4078$\pm$0.0019 | 0.3892$\pm$0.0021 | 0.4030$\pm$0.0016 | 0.3850$\pm$0.0020 | 0.3990$\pm$0.0015 |
| ETTm2 | 0.2817$\pm$0.0020 | 0.3265$\pm$0.0016 | 0.2742$\pm$0.0018 | 0.3199$\pm$0.0014 | 0.2699$\pm$0.0016 | 0.3161$\pm$0.0012 | 0.2670$\pm$0.0010 | 0.3130$\pm$0.0008 |
| Traffic | 0.4653$\pm$0.0043 | 0.2879$\pm$0.0027 | 0.4529$\pm$0.0039 | 0.2821$\pm$0.0024 | 0.4459$\pm$0.0033 | 0.2788$\pm$0.0019 | 0.4410$\pm$0.0019 | 0.2760$\pm$0.0014 |
| Electricity | 0.1772$\pm$0.0020 | 0.2743$\pm$0.0015 | 0.1725$\pm$0.0017 | 0.2688$\pm$0.0012 | 0.1698$\pm$0.0015 | 0.2656$\pm$0.0011 | 0.1680$\pm$0.0013 | 0.2630$\pm$0.0011 |
| Weather | 0.2553$\pm$0.0020 | 0.2816$\pm$0.0015 | 0.2485$\pm$0.0017 | 0.2759$\pm$0.0012 | 0.2447$\pm$0.0014 | 0.2727$\pm$0.0011 | 0.2420$\pm$0.0010 | 0.2700$\pm$0.0009 |
| Exchange | 0.3756$\pm$0.0038 | 0.4182$\pm$0.0026 | 0.3656$\pm$0.0034 | 0.4098$\pm$0.0022 | 0.3599$\pm$0.0028 | 0.4050$\pm$0.0018 | 0.3560$\pm$0.0016 | 0.4010$\pm$0.0010 |
| Avg. | 0.3492 (+5.50%) | 0.3566 (+4.30%) | 0.3399 (+2.70%) | 0.3494 (+2.20%) | 0.3346 (+1.10%) | 0.3453 (+1.00%) | 0.3310 | 0.3419 |

didate neurons generated from overlapping feature subsets may become highly correlated, resulting in redundant representations and unstable closed-form estimation. The selection strategy removes low-utility or duplicated candidates, while column dropout encourages more diverse feature usage. Together, these mechanisms improve the stability and generalization of forward structural learning.

**Ablation of Fuzzy Rule Gating.** Disabling fuzzy rule gating produces smaller but still consistent degradation. This indicates that rule-level local modeling contributes to performance, although its effect is less dominant than the Fourier-enhanced consequent. The fuzzy rule gates allow FIPN to decompose the global forecasting function into locally valid mappings under different temporal regimes. Without such routing, the model relies on a more global polynomial mapping, which weakens its ability to adapt to heterogeneous or non-stationary dynamics.

**Ablation of the Regularized Closed-Form Estimation.** Figure 2 examines the role of $\ell_2$-regularized closed-form estimation. On Traffic and Electricity, $\ell_2$ regularization consistently reduces MSE across all horizons, with more visible gains at longer prediction lengths. For example, $\ell_2$ regularization reduces the MSE from 0.403 to 0.392 on Traffic and from 0.141 to 0.137 on Electricity at $H$=96, with larger gains observed at $H$=720. The shaded bars further show that regularization provides stable gains when Fourier-enhanced consequents are used. This indicates that $\ell_2$ regularization helps stabilize correlated Fourier-polynomial design matrices by controlling coefficient variance and mitigating ill-conditioning.

### 4.3. Efficiency Analysis

We evaluate the efficiency of FIPN on ETTh1 under the long-horizon setting with look-back length $L = 96$, prediction horizon $H = 720$, and batch size 16. Four complementary metrics are reported: parameter count, MACs, peak memory, and training time, which respectively reflect model size, computational cost, memory footprint, and runtime efficiency. Detailed statistics are provided in Appendix Table 5.

Figure 3 presents the accuracy–efficiency trade-offs. As shown in Figure 3(a), FIPN lies on the Pareto frontier in the low-complexity region. It achieves competitive MSE with only 0.89K parameters, which is substantially smaller than DLinear, PatchTST, and Crossformer with 1.04M, 9.25M, and 17.40M parameters, respectively. This shows that FIPN can maintain strong predictive ability without relying on large-scale over-parameterization. Figure 3(b) further compares training time, MACs, and peak memory. FIPN remains in the Pareto-efficient region with the smallest overall footprint, requiring only 9.12M MACs, 26.01 MB peak memory, and 42.8 ms per iteration. Compared with Transformer-style and mixing-based baselines, FIPN avoids costly end-to-end backpropagation over deep architectures and reduces the need to store large activation and gradient buffers.

The efficiency advantage mainly comes from its forward self-organizing construction and $\ell_2$-regularized closed-form estimation. Useful polynomial nodes are selected layer by layer, while the compact Fourier-polynomial basis preserves frequency-aware modeling ability with only a small number of coefficients.

### 4.4. Hyperparameter Analysis

We analyze the sensitivity of FIPN to three key hyperparameters: the number of clusters $C$, the FCM fuzzification coefficient $\gamma$, and the Fourier band $[k_{\min}, k_{\max}]$. Figure 4 reports the averaged MSE on ETTh1, ETTm1, Traffic, and Exchange. Overall, FIPN exhibits clear but smooth sensitivity to these hyperparameters. The performance changes are observable when the hyperparameters vary, yet the curves do not fluctuate abruptly, suggesting that the model is tunable but not overly fragile within reasonable ranges. The optimal settings also differ slightly across datasets, which is

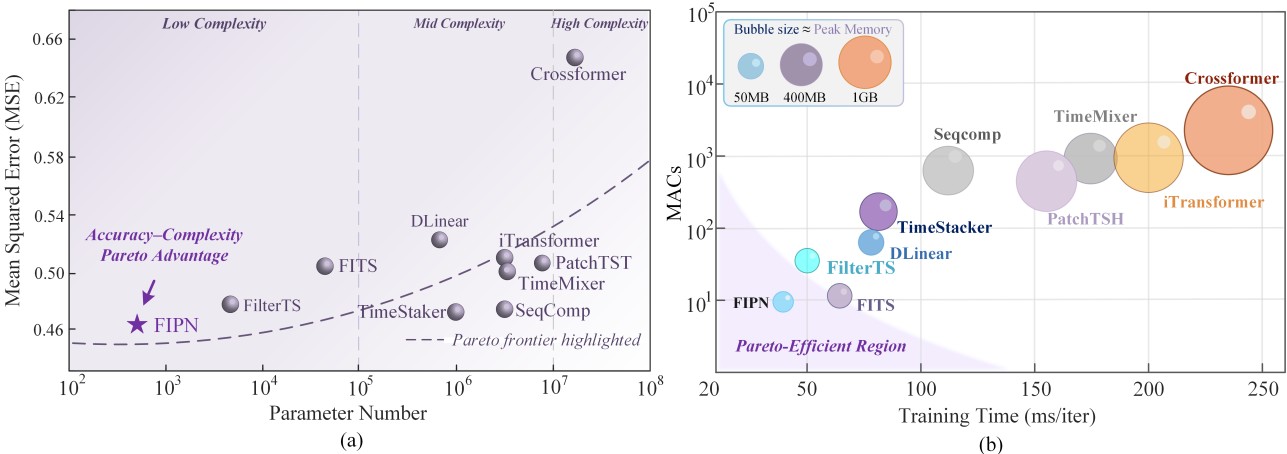

*Figure 3.* Accuracy-complexity and multi-objective efficiency trade-offs of FIPN. All results are reported on ETTh1 with lookback $L$=96, horizon $H$=720, and batch size 16. **(a)** Accuracy–complexity profile measured by MSE and parameter count, where the dashed curve denotes the Pareto frontier. **(b)** Multi-objective efficiency comparison in terms of training time, computational cost (MACs), and peak memory usage. Bubble size represents peak memory, and the shaded region indicates the Pareto-efficient regime.

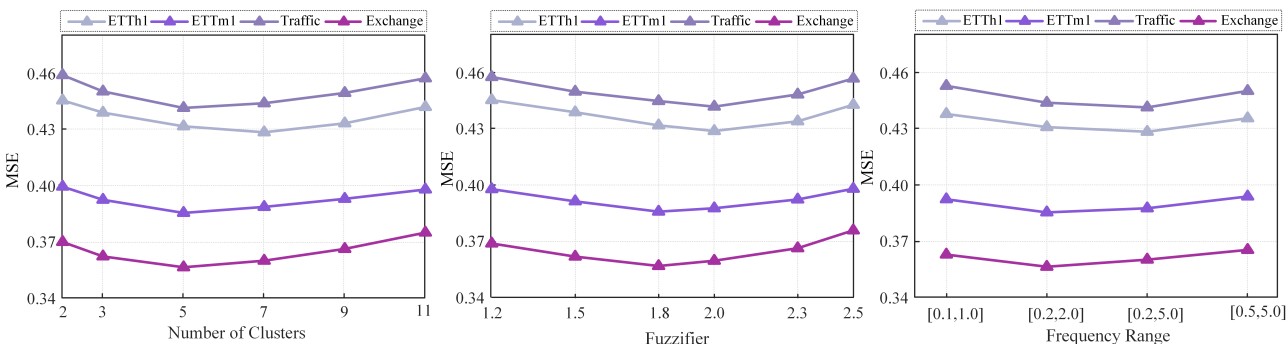

*Figure 4.* Hyperparameter sensitivity of FIPN with respect to the number of clusters, fuzzifier, and frequency range. The results are reported in terms of MSE on representative datasets.

expected because these benchmarks have different levels of periodicity, noise, and non-stationarity.

As $C$ increases within $\{2, 3, 5, 7, 9, 11\}$, the error generally decreases from under-parameterized settings, reaches the best performance at a moderate rule budget, and then saturates or slightly degrades. This suggests that rule gating requires enough local experts to capture heterogeneous temporal dynamics, but an overly large rule bank may over-fragment the input space and introduce redundant or highly correlated local models, thereby weakening closed-form estimation under forward growth. For $\gamma \in \{1.2, 1.5, 1.8, 2.0, 2.3, 2.5\}$, the most reliable region is around $\gamma \in [1.8, 2.0]$. A smaller $\gamma$ produces sharper and more boundary-sensitive memberships, increasing variance in rule activation, whereas a larger $\gamma$ over-smooths the memberships and weakens local specialization. For the Fourier band $[k_{\min}, k_{\max}]$, the results are relatively stable as long as low-to-middle frequency components are covered. Too narrow a band may miss useful seasonal or oscillatory patterns,

while an overly wide band may introduce high-frequency bases that are harder to estimate and more sensitive to noise. These observations suggest that $C$, $\gamma$, and $[k_{\min}, k_{\max}]$ serve as interpretable capacity controls for local-expert granularity, gating smoothness, and spectral coverage, respectively.

## 5. Conclusion, Limitations and Future Work

**Summary and Contributions.** This work studies long-horizon time-series forecasting from the perspective of efficiency and structural interpretability. We propose FIPN, a forward self-organizing polynomial network that builds its architecture layer by layer through candidate generation, closed-form estimation, and validation-based selection, without relying on backpropagation-based optimization. Each neuron combines a fuzzy rule-gated antecedent with a Fourier-enhanced polynomial consequent, enabling local regime interpretation and compact modeling of periodic temporal structures. To stabilize forward growth, FIPN

further incorporates $\ell_2$-regularized fitting, regularized node scoring, column dropout, and persistent access to raw inputs. Experiments on standard long-horizon forecasting benchmarks show that FIPN achieves competitive accuracy with a compact model size and low computational cost, supporting a practical balance among accuracy, efficiency, and structurally traceable interpretability.

**Current Limitations.** The current rule-gating mechanism in FIPN is driven by fuzzy clustering, and the Fourier-enhanced consequent adopts a fixed frequency range. These design choices may be suboptimal when underlying regimes or dominant spectral components evolve rapidly over time. In addition, the forward structure growth procedure is inherently greedy and may not always allocate model capacity optimally, particularly at greater depths. Finally, the present evaluation focuses on deterministic point forecasting.

**Future Directions.** Future work can extend FIPN in several directions. Adaptive spectral representations may be introduced to adjust frequency bases under non-stationary temporal patterns. More flexible gating mechanisms could also be developed to improve local regime partitioning. In addition, online or streaming variants of FIPN would be useful for real-time forecasting scenarios. Finally, extending the framework to probabilistic forecasting and evaluating its robustness under missing data and stronger distribution shifts remain important directions.

## Impact Statement

This paper presents work whose goal is to advance the field of Machine Learning. There are many potential societal consequences of our work, none which we feel must be specifically highlighted here.

## Acknowledgements

This work was supported in part by the Natural Science Foundation of Shandong Province under Grant No. ZR2025MS18, and by the Basic Science Research Program through the National Research Foundation of Korea (NRF), funded by the Ministry of Education, under Grant Nos. NRF-2022R1I1A1A01071671 and RS-2026-25474453.

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

# A. Datasets and Experimental Protocol

## A.1. Dataset Description

We conduct experiments on eight widely used real-world benchmarks for multivariate time-series forecasting. These datasets cover representative application domains, including electricity, transportation, weather, and finance. Their statistics are summarized in Table 3, and brief descriptions are given below.

- **ETT Dataset:** The ETT benchmark consists of four subsets collected from electricity transformers, including two hourly datasets (ETTh1 and ETTh2) and two 15-minute datasets (ETTm1 and ETTm2). Each subset contains load-related variables and the transformer oil temperature target, recorded from July 2016 to July 2018.

- **Traffic Dataset:** This dataset contains hourly road-occupancy measurements collected from a large-scale sensor network on San Francisco freeways. It spans 2015–2016 and contains clear daily and weekly periodic patterns as well as regime changes.

- **Electricity Dataset:** This dataset records hourly electricity consumption from 321 customers between 2012 and 2014. It is a representative high-dimensional benchmark for multivariate forecasting.

- **Weather Dataset:** This dataset contains 21 meteorological variables, such as air temperature, humidity, and related atmospheric indicators. The observations are sampled every 10 minutes throughout 2020, exhibiting strong seasonality and multi-scale dynamics.

- **Exchange Dataset:** This dataset contains daily exchange rates of eight currencies from 1990 to 2016. It is commonly used to evaluate forecasting robustness under noisy and non-stationary financial fluctuations.

*Table 3.* Dataset statistics. *Variates* denotes the number of variables, *Dataset Size* denotes the total number of time points, *Frequency* denotes the sampling interval, and *Information* denotes the application domain.

| Dataset | ETTh1 | ETTh2 | ETTm1 | ETTm2 | Traffic | Electricity | Weather | Exchange |
|---|---|---|---|---|---|---|---|---|
| **Variates** | 7 | 7 | 7 | 7 | 862 | 321 | 21 | 8 |
| **Dataset Size** | 14,307 | 14,307 | 57,500 | 57,500 | 17,451 | 26,321 | 52,603 | 7,207 |
| **Frequency** | Hourly | Hourly | 15min | 15min | Hourly | Hourly | 10min | Daily |
| **Information** | Electricity | Electricity | Electricity | Electricity | Transportation | Electricity | Weather | Finance |

## A.2. Evaluation Metrics and Data Splits

To evaluate long-horizon forecasting performance across datasets, we adopt Mean Absolute Error (MAE) and Mean Squared Error (MSE) as the primary metrics. These two metrics characterize prediction errors from complementary perspectives.

**Problem setup and sliding-window protocol.** Let $\mathbf{X} \in \mathbb{R}^{R \times T}$ denote a multivariate time series with $R$ channels and $T$ time steps. Given a look-back window of length $L$, the goal is to predict the next $H$ steps. Under the sliding-window protocol, at time index $t$, the input and target are defined as

$$\mathbf{X}_t = \mathbf{X}_{:,\,t-L+1:t} \in \mathbb{R}^{R \times L}, \quad \mathbf{Y}_t = \mathbf{X}_{:,\,t+1:t+H} \in \mathbb{R}^{R \times H}, \tag{15}$$

respectively. The model prediction is denoted by $\widehat{\mathbf{Y}}_t \in \mathbb{R}^{R \times H}$.

$$\text{MAE} = \frac{1}{RH} \sum_{r=1}^{R} \sum_{h=1}^{H} \left| Y_t(r,h) - \widehat{Y}_t(r,h) \right|. \tag{16}$$

$$\text{MSE} = \frac{1}{RH} \sum_{r=1}^{R} \sum_{h=1}^{H} \left( Y_t(r,h) - \widehat{Y}_t(r,h) \right)^2. \tag{17}$$

Here, $Y_t(r, h)$ and $\widehat{Y}_t(r, h)$ denote the ground-truth and predicted values at channel $r$ and forecasting step $h$, respectively. MSE penalizes large deviations more heavily due to the squared term, while MAE measures the average absolute deviation and is generally less sensitive to outliers. Together, they provide complementary views of forecasting accuracy.

**Avg Improve (%).**  To summarize ablation effects across datasets, we compute the relative improvement of the full model over each ablated variant for dataset $d$:

$$\text{Improve}_d = \frac{E_d^{(\text{abl})} - E_d^{(\text{full})}}{E_d^{(\text{abl})}} \times 100\%, \tag{18}$$

where $E$ denotes either MSE or MAE, computed separately. The reported Avg Improve is obtained by averaging $\text{Improve}_d$ over all datasets.

# B. Implementation Details

## B.1. Baseline Descriptions

We compare FIPN with representative forecasting methods spanning lightweight linear baselines, Transformer-style architectures, and recent efficient long-horizon forecasters. All baselines are evaluated under the same data splits, input length, and forecasting horizons for fair comparison.

**DLinear.** DLinear is a lightweight linear forecasting model that decomposes a time series into trend and seasonal components and applies linear projections to each part. It serves as a strong efficiency-oriented baseline.

**Crossformer.** Crossformer is a Transformer-based model that constructs segment-wise representations and uses cross-dimension attention to capture long-range temporal dependencies and inter-variable interactions.

**PatchTST.** PatchTST is a patch-based Transformer that divides each variable history into local temporal patches and applies self-attention over patch tokens, improving scalability for long-horizon forecasting.

**iTransformer.** iTransformer is an inverted Transformer variant that treats variables as tokens, allowing the model to better capture cross-variable dependencies while retaining the expressive power of attention mechanisms.

**TimeMixer.** TimeMixer is a multiscale mixing model that decomposes temporal patterns across different scales and integrates complementary information for non-stationary time-series forecasting.

**SeqComp.** SeqComp is a sequence complementor framework that introduces learnable complementary sequences to enrich temporal representations and improve long-horizon forecasting.

**FilterTS.** FilterTS is a frequency-filtering-based forecasting model that uses compact filtering operations to capture useful temporal and spectral patterns in multivariate time series.

**FITS.** FITS is a compact frequency-domain forecasting model that uses complex-valued interpolation to capture temporal variations with a small parameter budget, serving as an efficient spectral baseline.

**TimeStacker.** TimeStacker is a multilevel observation framework that stacks temporal representations at different levels to better capture non-stationary patterns and regime changes.

## B.2. Complexity Analysis

We analyze the computational complexity of FIPN with respect to the number of samples $N$, input feature dimension $d$, number of fuzzy rules $C$, maximum harmonic order $r_{\max}$, number of evaluated candidates per layer $M$, and maximum forward depth $P$. At each layer, FIPN generates a candidate pool, fits each candidate independently by closed-form ridge regression, and retains a compact subset through validation-based selection.

For a candidate neuron, fuzzy membership computation requires distance evaluation with respect to $C$ prototypes, leading to a cost of $\mathcal{O}(NCd)$ per FCM iteration. Since the number of FCM iterations is fixed in practice, it is treated as a constant factor. The Fourier-enhanced consequent then constructs rule-wise harmonic features. With bounded $C$ and $r_{\max}$, the number of consequent features $D_j$ is also bounded, and the feature construction cost is linear in $N$.

Given the design matrix $\mathbf{F}_j \in \mathbb{R}^{N \times D_j}$, ridge-regularized closed-form estimation has a cost of $\mathcal{O}(ND_j^2 + D_j^3)$. Because $D_j$

is small and bounded in FIPN, this cost scales linearly with the sample size, i.e., $\mathcal{O}(N)$ up to constant factors. If at most $M$ candidates are evaluated at each layer, the per-layer cost is $\mathcal{O}(MN)$, and the overall training complexity over $P$ forward layers is $\mathcal{O}(PMN)$ under bounded width and consequent dimension.

Unlike attention-based forecasters that rely on iterative backpropagation and dense temporal interactions, FIPN performs forward candidate fitting and validation-based selection with closed-form estimation. This substantially reduces the training overhead and memory requirement associated with gradient backpropagation. Table 4 summarizes the complexity of FIPN and representative forecasting baselines.

*Table 4.* Complexity comparison of representative time-series forecasters. Here, $N$ denotes the number of samples, $T$ the sequence length, $d$ the feature dimension, $h$ the hidden dimension, $P$ the maximum forward depth, $M$ the number of evaluated candidates per layer, and $K$ the bounded number of Fourier-polynomial basis terms.

| Model | Training Complexity | Inference Complexity | Optimization | Interpretability |
|---|---|---|---|---|
| LSTM | $\mathcal{O}(NTh^2)$ | $\mathcal{O}(Th^2)$ | Backpropagation | Low |
| Transformer | $\mathcal{O}(NT^2d)$ | $\mathcal{O}(T^2d)$ | Backpropagation | Low |
| Autoformer | $\mathcal{O}(NT\log T)$ | $\mathcal{O}(T\log T)$ | Backpropagation | Medium |
| DLinear | $\mathcal{O}(NTd)$ | $\mathcal{O}(Td)$ | Closed-form | Low |
| FIPN (ours) | $\mathcal{O}(PMN)$ | $\mathcal{O}(PdK)$ | Forward closed-form | High |

## B.3. Efficiency Analysis

Table 5 provides the detailed resource measurements for the efficiency comparison reported in the main paper. We evaluate each method under the same experimental setting and record four complementary indicators: parameter count (**Params**), computational cost (**MACs**), peak training memory usage (**Peak Mem**), and per-iteration training time (**Train Time**). These metrics jointly characterize the practical resource footprint of a forecaster, covering model size, arithmetic intensity, memory pressure, and wall-clock speed. To improve comparability, MACs and Train Time are computed under identical input/output lengths (look-back window $L$ and forecast horizon $H$) and a fixed batch size, while Peak Mem corresponds to the maximum allocated memory observed during training.

As shown in Table 5, FIPN exhibits a particularly lightweight footprint across all four dimensions, with substantially fewer parameters and lower compute/memory demand than representative deep baselines. This efficiency comes from the forward self-organizing construction and ridge-regularized closed-form estimation, which avoid iterative backpropagation through deep temporal blocks. At the same time, the compact Fourier-polynomial basis preserves frequency modeling ability with only a small number of coefficients.

*Table 5.* Detailed efficiency statistics on ETTh1 with look-back window $L = 96$, forecast horizon $H = 720$, and batch size 16.

| Method | Params | MACs | Peak Mem (MB) | Train Time (ms) |
|---|---|---|---|---|
| DLinear | 1.04M | 89.99M | 64.11 | 76.8 |
| Crossformer | 17.40M | 915M | 1474.56 | 238.5 |
| PatchTST | 9.25M | 580M | 420.93 | 157.2 |
| iTransformer | 5.47M | 730M | 356.12 | 198.7 |
| TimeMixer | 5.58M | 657M | 250.10 | 176.4 |
| SeqComp | 5.49M | 720M | 155.88 | 112.3 |
| TimeStacker | 1.39M | 120M | 87.47 | 84.5 |
| FITS | 79.15K | 14.62M | 74.82 | 63.5 |
| FilterTS | 7.12K | 57.23M | 76.80 | 51.5 |
| FIPN | 0.89K | 9.12M | 26.01 | 42.8 |

## B.4. Layer-wise Utility Analysis

We provide a layer-wise utility analysis in Table 6 to examine whether deeper forward-grown polynomial networks consistently improve generalization or quickly lead to redundant nodes and overfitting. Here, PNN and GMDH are classical greedy layer-wise polynomial networks (Tetko et al., 2000; Ivakhnenko, 1971), while PFPNN is a recent progressive fuzzy variant with gated selection (Wang et al., 2023). We follow the same ETTh1 protocol as in the main experiments and record

the MSE/MAE at each depth using each method's standard forward selection rule.

The results show that classical PNN and GMDH deteriorate after only a few layers. PNN improves until Layer 3 but degrades at Layer 4, and GMDH shows a similar trend. PFPNN benefits from moderate depth and reaches its best performance around Layer 4, but becomes unstable when the network grows deeper. These results indicate that naive greedy forward growth can easily introduce redundant or highly correlated nodes, so additional depth does not necessarily provide useful information and may even amplify overfitting or numerical instability.

In contrast, FIPN exhibits a more stable depth-wise behavior. Its performance improves gradually from shallow layers and reaches the best result around Layer 5, which is consistent with the ETTh1 result reported in the main table. After that, the MSE/MAE remain within a narrow range, suggesting that the effective depth of FIPN saturates around Layer 5. Further growth only introduces mild fluctuations rather than severe degradation.

This stability mainly comes from two design choices. First, persistent raw-input access preserves feature diversity and prevents deeper layers from relying only on increasingly similar intermediate nodes. Second, regularized node scoring and column dropout suppress redundant candidates and reduce multicollinearity during forward selection. Therefore, Table 6 complements the main results by showing that FIPN not only achieves competitive accuracy, but also maintains a more stable depth–utility trade-off than classical forward-grown polynomial baselines.

*Table 6.* Layer-wise utility comparison of PNN, GMDH, PFPNN, and FIPN on ETTh1. Lower MSE and MAE indicate better performance. "N/A" denotes that the corresponding method fails to yield a stable or valid deeper model under its standard forward selection rule.

| Layer | PNN | | GMDH | | PFPNN | | FIPN | |
|---|---|---|---|---|---|---|---|---|
| | MSE | MAE | MSE | MAE | MSE | MAE | MSE | MAE |
| 1 | 14.627 | 5.911 | 15.104 | 6.742 | 5.824 | 3.601 | 0.514 | 0.470 |
| 2 | 10.739 | 5.571 | 11.731 | 5.724 | 3.630 | 2.208 | 0.471 | 0.444 |
| 3 | 9.403 | 5.333 | 9.665 | 5.461 | 1.824 | 1.991 | 0.451 | 0.433 |
| 4 | 12.184 | 6.284 | 12.936 | 6.617 | 1.486 | 0.972 | 0.436 | 0.426 |
| 5 | N/A | N/A | N/A | N/A | 2.184 | 1.298 | 0.428 | 0.422 |
| 6 | N/A | N/A | N/A | N/A | 5.913 | 4.164 | 0.429 | 0.422 |
| 7 | N/A | N/A | N/A | N/A | 7.814 | 6.829 | 0.430 | 0.423 |
| 8 | N/A | N/A | N/A | N/A | N/A | N/A | 0.431 | 0.423 |
| 9 | N/A | N/A | N/A | N/A | N/A | N/A | 0.433 | 0.424 |
| 10 | N/A | N/A | N/A | N/A | N/A | N/A | 0.435 | 0.425 |

### B.5. Horizon-wise Hyperparameter Sensitivity

To complement the horizon-averaged analysis in the main text, we further report horizon-wise hyperparameter sensitivity in Figure 5. The results are shown separately for each prediction horizon $H \in \{96, 192, 336, 720\}$. This breakdown verifies that the conclusions drawn from the averaged results are not dominated by a single horizon. Across datasets and horizons, varying the number of clusters $C$, the fuzzification coefficient $\gamma$, and the Fourier band $[k_{\min}, k_{\max}]$ leads to observable changes in forecasting accuracy, indicating that FIPN is responsive to even small hyperparameter variations. Nevertheless, the overall trends remain smooth and largely consistent. Moderate settings usually provide better or comparable performance, whereas overly small or large values may degrade the results. This reflects the trade-offs among regime partition granularity, gating smoothness, and spectral coverage. Overall, the horizon-wise curves show that the selected hyperparameter ranges are effective, while also confirming the necessity of careful tuning for stable short- and long-horizon forecasting.

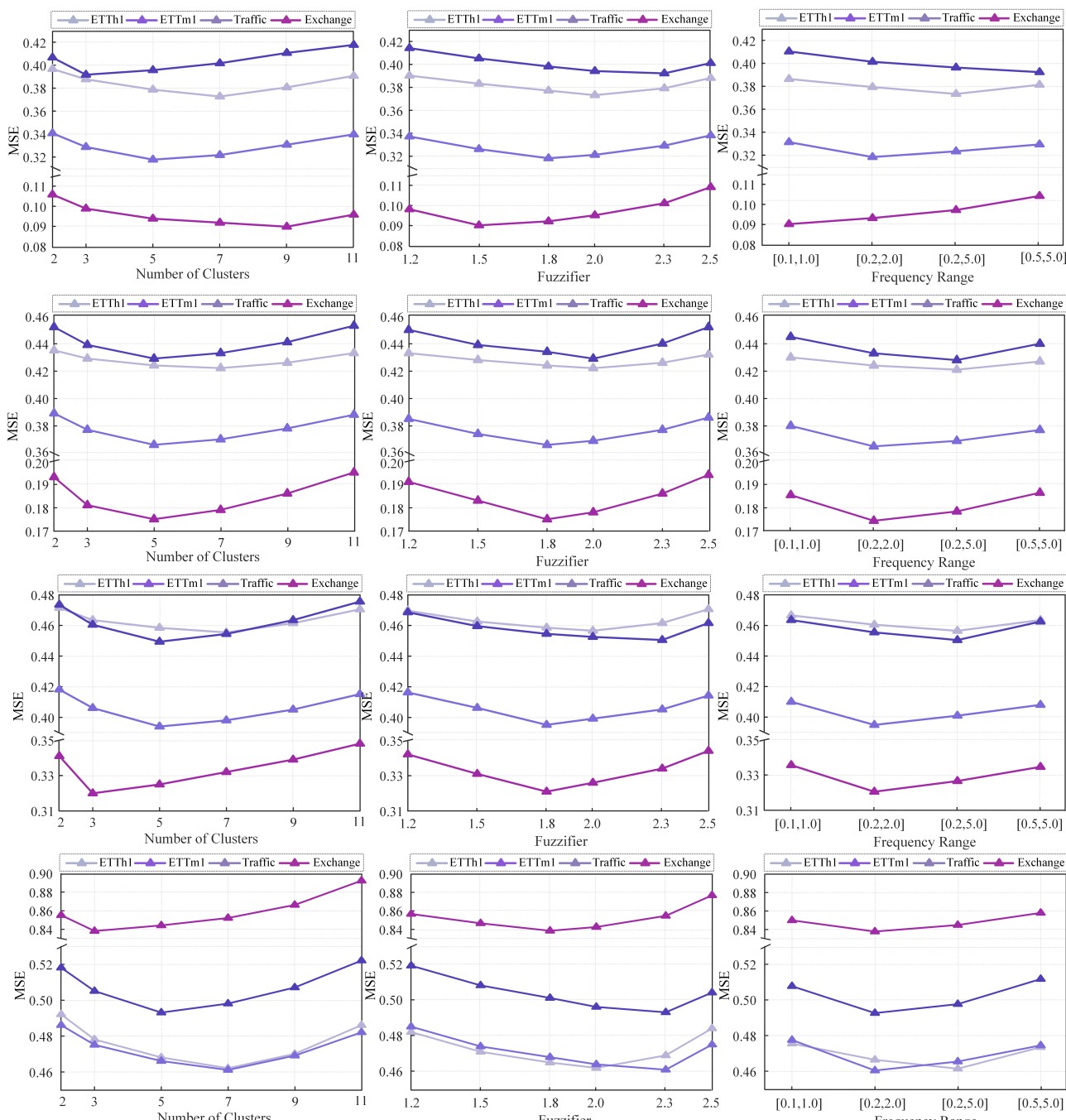

*Figure 5.* Horizon-wise hyperparameter sensitivity of FIPN. We report MSE for each prediction horizon $H \in \{96, 192, 336, 720\}$ on representative datasets, including ETTh1, ETTm1, Traffic, and Exchange. The sensitivity is evaluated with respect to the number of clusters $C$, the fuzzification coefficient $\gamma$, and the Fourier band $[k_{\min}, k_{\max}]$. Lower values indicate better performance.

## B.6. Rule-Interpretable Neuron Trace-back and Activation Analysis

To further illustrate the structural interpretability of FIPN, we provide a concrete trace-back example on ETTh1 under the $L = 96$, $H = 96$ setting. The purpose of this analysis is not to assign direct physical semantics to every learned prototype, but to show that the final predictor can be inspected through its layer-wise formation, fuzzy rule activations, and local Fourier-polynomial consequents.

In this example, the selected neuron is $N_{396}^{(5)}$ at Layer 5. Its dependency path can be recursively traced as

$$N_{396}^{(5)} \leftarrow \left\{ N_{18}^{(4)}, N_{35}^{(4)}(x_5) \right\},$$

where $N_{35}^{(4)}(x_5)$ denotes a terminal branch associated with raw input $x_5$. The remaining branch is expanded as

$$N_{18}^{(4)} \leftarrow \left\{ N_{27}^{(3)}, N_{33}^{(3)}(x_3) \right\},$$

where $N_{33}^{(3)}(x_3)$ is another terminal branch associated with raw input $x_3$. Further expanding $N_{27}^{(3)}$ gives

$$N_{27}^{(3)} \leftarrow \left\{ N_9^{(2)}, N_{14}^{(2)} \right\},$$

with

$$N_9^{(2)} \leftarrow \left\{ N_4^{(1)}, N_8^{(1)} \right\}, \qquad N_{14}^{(2)} \leftarrow \left\{ N_2^{(1)}, N_{11}^{(1)} \right\}.$$

The corresponding Layer-1 neurons are generated from raw input pairs:

$$N_4^{(1)} \leftarrow (x_1, x_5), \qquad N_8^{(1)} \leftarrow (x_2, x_5),$$
$$N_2^{(1)} \leftarrow (x_1, x_3), \qquad N_{11}^{(1)} \leftarrow (x_3, x_5).$$

Thus, $N_{396}^{(5)}$ can be unfolded back to earlier-layer nodes and raw input pairs rather than being treated as an opaque hidden unit. The repeated appearance of $x_3$ and $x_5$ along the selected path also suggests that these raw inputs are consistently retained as informative ancestors during forward growth.

We next examine the rule-gated local form of the selected neuron. Following the notation in Section 3.1, fuzzy $C$-means produces memberships $u_{nq}$ and prototypes $v_q$, where $q \in \{1, \ldots, C\}$. The normalized rule gate for sample $z_n$ is

$$\beta_{nq} = \frac{u_{nq}^\gamma}{\sum_{r=1}^C u_{nr}^\gamma}, \qquad \sum_{q=1}^C \beta_{nq} = 1.$$

For $N_{396}^{(5)}$, we use $C = 7$ fuzzy rules, fuzzification coefficient $\gamma = 2.0$, and Fourier band $[0.2, 2.0]$. Let $z_n \in \mathbb{R}^2$ denote the two-dimensional input representation received by this selected neuron after trace-back composition. The neuron output is written as a rule-weighted mixture of local Fourier-polynomial consequents:

$$h_{396}^{(5)}(z_n) = \sum_{q=1}^7 \beta_{nq} \, g_q(z_n),$$

where $g_q(\cdot)$ denotes the consequent of rule $q$. A compact local form is

$$g_q(z_n) = a_{q,0} + \sum_{i=1}^2 \sum_{r=1}^{r_{\max}} a_{q,i,r} \, \psi_{q,i,r}(z_n),$$

with

$$\psi_{q,i,r}(z_n) = \cos(\omega_{q,i,r}(z_{n,i} - v_{q,i}) + \phi_{q,i,r}).$$

Here, $v_{q,i}$ is the $i$-th component of prototype $v_q$, $\omega_{q,i,r}$ is the rule- and dimension-specific angular frequency, $\phi_{q,i,r}$ is the phase offset, and $a_{q,i,r}$ is the closed-form estimated consequent coefficient. Therefore, each rule defines a local Fourier-polynomial expert, while $\beta_{nq}$ determines how strongly this expert contributes to the neuron output for sample $z_n$.

Figure 6 visualizes the trace-back structure and the normalized rule-gate activations of $N_{396}^{(5)}$. In the activation heatmap, each row corresponds to one fuzzy rule and each column corresponds to a representative sample or temporal position. Warmer colors indicate larger $\beta_{nq}$, meaning that the corresponding local expert contributes more strongly. The heatmap shows that different rules dominate in different temporal regions, suggesting that the selected neuron does not rely on a single global mapping. Instead, it softly switches among multiple local Fourier-polynomial experts according to the inferred input regime.

Overall, the trace-back graph and rule activation heatmap provide complementary views of interpretability. The trace-back graph explains how the selected neuron is structurally composed from earlier-layer nodes and raw input pairs, while the activation heatmap shows how its fuzzy antecedent assigns samples to local experts. Together with the explicit Fourier-polynomial consequent, this gives an inspectable view of FIPN in terms of layer-wise formation, rule activation, and local functional behavior.

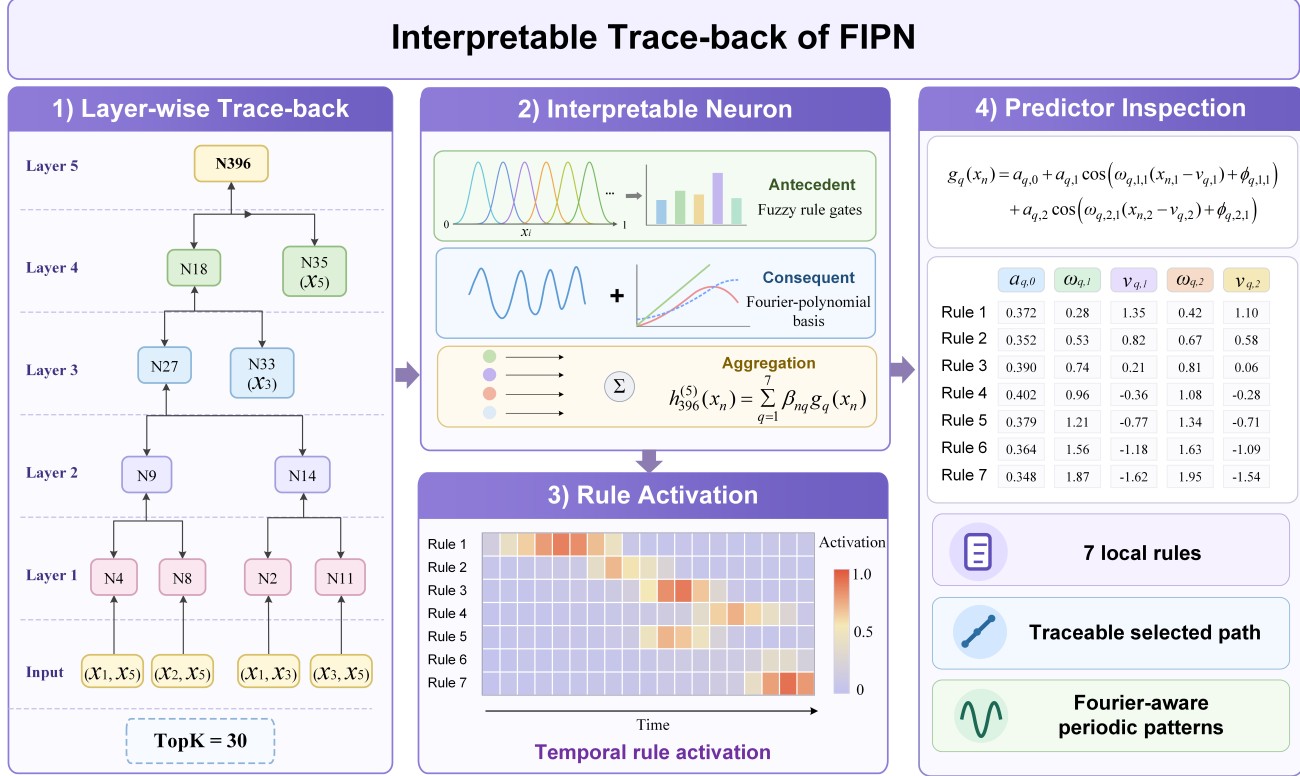

*Figure 6.* Interpretable trace-back and rule-gated explanation of the selected neuron $N_{396}^{(5)}$ on ETTh1 under the $L = 96, H = 96$ setting. The left panel shows the layer-wise dependency path, where the selected Layer-5 neuron is recursively traced back to earlier-layer nodes and raw input pairs. The middle panels decompose the neuron into an FCM-based fuzzy antecedent and a Fourier-enhanced polynomial consequent, and visualize the normalized rule-gate activations $\beta_{nq}$ across representative samples. The right panel summarizes the local Fourier-polynomial consequent form, illustrating that the final predictor can be inspected through structural formation, rule activation, and local expert behavior.

### B.7. Comparison with Non-gradient Baselines

We further compare FIPN with two representative non-gradient forecasting baselines in Table 7: LCESN, an ESN-style reservoir model with local connectivity, and PFPNN, a progressive fuzzy polynomial network based on forward growth. This comparison provides a closer family-level reference to the proposed forward-grown and closed-form learning paradigm.

As shown in Table 7, FIPN achieves lower errors on most benchmarks under the same evaluation protocol. This suggests that the proposed design is not only competitive with gradient-trained forecasters in the main text, but also compares favorably with representative non-backpropagation models. Compared with LCESN, which mainly relies on a fixed reservoir representation and a trained readout, FIPN explicitly constructs rule-gated Fourier-polynomial nodes through forward selection. Compared with PFPNN, FIPN further introduces Fourier-enhanced consequents and redundancy-controlled node selection, which help stabilize deeper forward construction and reduce sensitivity to correlated candidate nodes. Overall, these results support the effectiveness of combining fuzzy rule gating, spectral modeling, and closed-form node selection within a unified forward-grown forecasting framework.

### B.8. Visualization of Forecasting Results

As shown in Figure 7, FIPN better preserves long-horizon temporal dynamics compared with representative baseline methods. It follows the ground-truth trajectory more closely, reduces phase drift, and avoids excessive smoothing. This qualitative observation is consistent with the quantitative results reported in the main experiments.

*Table 7.* Comparison with representative non-gradient forecasting baselines on eight benchmarks. Results are reported as horizon-averaged MSE/MAE, where lower values indicate better performance.

| Dataset | FIPN | LCESN | PFPNN |
|---------|------|-------|-------|
| ETTh1 | 0.428/0.422 | 0.818/0.662 | 1.486/0.972 |
| ETTh2 | 0.361/0.391 | 0.438/0.464 | 1.372/0.911 |
| ETTm1 | 0.385/0.399 | 0.412/0.430 | 1.528/0.986 |
| ETTm2 | 0.267/0.313 | 0.282/0.354 | 1.214/0.846 |
| Traffic | 0.441/0.276 | 0.882/0.407 | 1.968/1.142 |
| Electricity | 0.168/0.263 | 0.225/0.329 | 1.108/0.801 |
| Weather | 0.242/0.270 | 0.232/0.279 | 1.296/0.887 |
| Exchange | 0.356/0.401 | 0.889/0.661 | 1.842/1.306 |

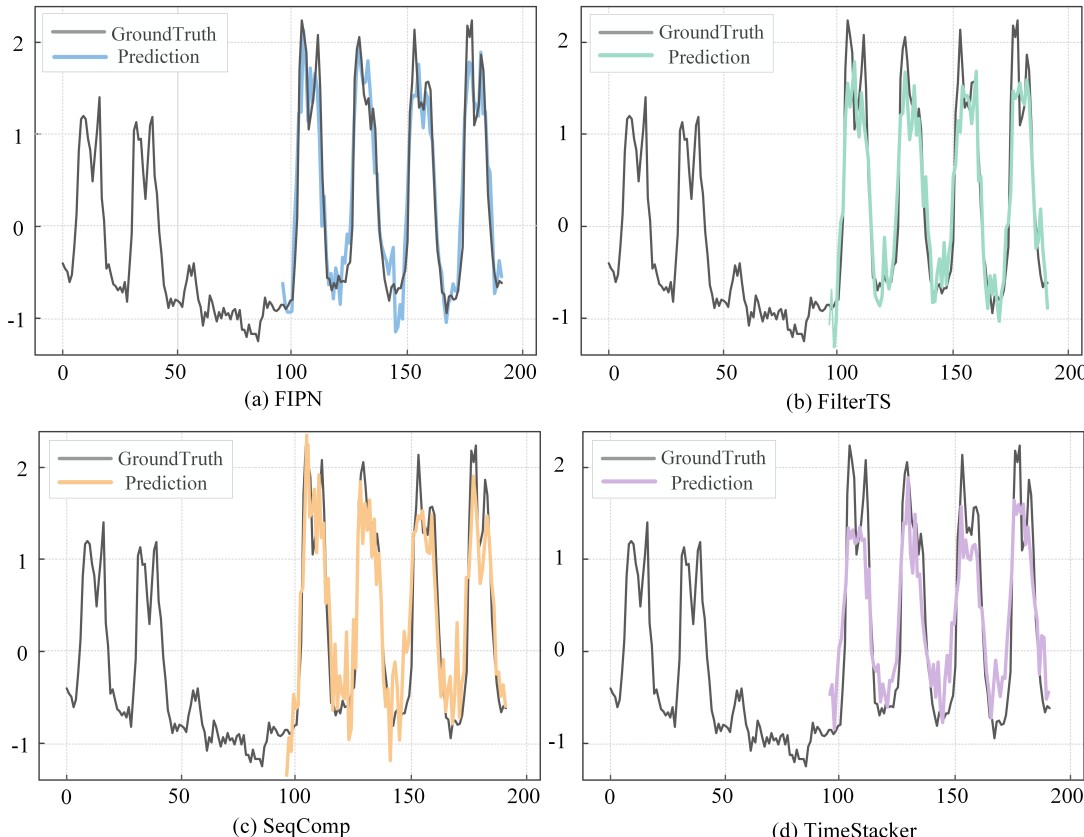

*Figure 7.* Qualitative 96-step-ahead forecasting results on the Electricity dataset ($L = 96$, $H = 96$).

# C. Full Results

Table 8 reports the complete horizon-wise forecasting results on eight real-world benchmarks, including ETTh1, ETTh2, ETTm1, ETTm2, Traffic, Electricity, Weather, and Exchange. Following the standard protocol, we fix the look-back length to $L = 96$ and evaluate four forecasting horizons $H \in \{96, 192, 336, 720\}$ using MSE and MAE.

For each dataset, the "Avg." row reports the mean error over the four horizons, providing an overall summary of long-horizon forecasting performance. The best results are highlighted in bold, and the second-best results are underlined. These full results complement the horizon-averaged comparison in the main text and provide a finer-grained view of model behavior as the forecasting horizon increases.

*Table 8.* Full results of the long-term multivariate forecasting task.

| Models | Horizon | FIPN (ours) | | TimeStacker (2025) | | FilterTS (2025) | | SeqComp (2025) | | TimeMixer (2024) | | FITS (2024) | | iTransformer (2024) | | PatchTST (2023) | | Crossformer (2023) | | DLinear (2023) | |
|---|---|---|---|---|---|---|---|---|---|---|---|---|---|---|---|---|---|---|---|---|---|
| | | MSE | MAE | MSE | MAE | MSE | MAE | MSE | MAE | MSE | MAE | MSE | MAE | MSE | MAE | MSE | MAE | MSE | MAE | MSE | MAE |
| ETTh1 | 96 | **0.373** | **0.383** | 0.379 | 0.385 | 0.374 | 0.391 | 0.375 | 0.394 | 0.375 | 0.400 | 0.386 | 0.396 | 0.386 | 0.405 | 0.414 | 0.419 | 0.423 | 0.448 | 0.386 | 0.400 |
| | 192 | **0.422** | **0.413** | 0.429 | 0.416 | 0.424 | 0.421 | 0.423 | 0.425 | 0.429 | 0.421 | 0.436 | 0.423 | 0.441 | 0.436 | 0.460 | 0.445 | 0.471 | 0.474 | 0.437 | 0.432 |
| | 336 | **0.456** | **0.436** | 0.459 | **0.436** | 0.464 | 0.441 | 0.457 | 0.448 | 0.484 | 0.458 | 0.478 | 0.444 | 0.487 | 0.458 | 0.501 | 0.466 | 0.570 | 0.546 | 0.481 | 0.459 |
| | 720 | **0.462** | 0.456 | 0.464 | **0.455** | 0.470 | 0.466 | **0.462** | 0.472 | 0.498 | 0.482 | 0.502 | 0.495 | 0.503 | 0.491 | 0.500 | 0.488 | 0.653 | 0.621 | 0.519 | 0.516 |
| | AVG | **0.428** | **0.422** | 0.433 | 0.423 | 0.433 | 0.430 | 0.429 | 0.435 | 0.447 | 0.440 | 0.451 | 0.440 | 0.454 | 0.447 | 0.469 | 0.455 | 0.529 | 0.522 | 0.456 | 0.452 |
| ETTh2 | 96 | **0.278** | 0.337 | 0.280 | **0.327** | 0.290 | 0.338 | 0.283 | 0.337 | 0.289 | 0.341 | 0.295 | 0.350 | 0.297 | 0.349 | 0.302 | 0.348 | 0.745 | 0.584 | 0.333 | 0.387 |
| | 192 | 0.366 | **0.384** | 0.373 | 0.385 | 0.374 | 0.390 | **0.362** | 0.388 | 0.372 | 0.392 | 0.381 | 0.396 | 0.380 | 0.400 | 0.388 | 0.400 | 0.877 | 0.656 | 0.477 | 0.476 |
| | 336 | 0.391 | 0.419 | 0.407 | 0.416 | 0.406 | 0.420 | 0.408 | 0.425 | **0.386** | **0.414** | 0.426 | 0.438 | 0.428 | 0.432 | 0.426 | 0.433 | 1.043 | 0.731 | 0.594 | 0.541 |
| | 720 | **0.409** | **0.424** | 0.412 | 0.431 | 0.418 | 0.437 | 0.419 | 0.441 | 0.412 | 0.434 | 0.431 | 0.446 | 0.427 | 0.445 | 0.431 | 0.446 | 1.104 | 0.763 | 0.831 | 0.657 |
| | AVG | **0.361** | 0.391 | 0.368 | **0.390** | 0.372 | 0.396 | 0.368 | 0.398 | 0.364 | 0.395 | 0.383 | 0.408 | 0.383 | 0.407 | 0.387 | 0.407 | 0.942 | 0.684 | 0.559 | 0.515 |
| ETTm1 | 96 | 0.318 | 0.351 | **0.311** | **0.337** | 0.321 | 0.360 | 0.321 | 0.359 | 0.320 | 0.357 | 0.355 | 0.375 | 0.334 | 0.368 | 0.329 | 0.367 | 0.404 | 0.426 | 0.345 | 0.372 |
| | 192 | 0.366 | 0.382 | 0.364 | **0.367** | 0.363 | 0.382 | 0.362 | 0.386 | **0.361** | 0.381 | 0.392 | 0.393 | 0.377 | 0.391 | 0.367 | 0.385 | 0.450 | 0.451 | 0.380 | 0.389 |
| | 336 | 0.395 | 0.406 | **0.389** | **0.391** | 0.395 | 0.403 | 0.393 | 0.406 | 0.390 | 0.404 | 0.424 | 0.414 | 0.426 | 0.420 | 0.399 | 0.410 | 0.532 | 0.515 | 0.413 | 0.413 |
| | 720 | 0.461 | 0.456 | 0.460 | **0.428** | 0.462 | 0.438 | **0.450** | 0.442 | 0.454 | 0.441 | 0.487 | 0.449 | 0.491 | 0.459 | 0.454 | 0.439 | 0.666 | 0.589 | 0.474 | 0.453 |
| | AVG | 0.385 | 0.399 | **0.381** | **0.381** | 0.385 | 0.396 | **0.381** | 0.398 | **0.381** | 0.395 | 0.415 | 0.408 | 0.407 | 0.410 | 0.387 | 0.400 | 0.496 | 0.496 | 0.403 | 0.407 |
| ETTm2 | 96 | **0.163** | **0.244** | 0.171 | 0.250 | 0.171 | 0.255 | 0.172 | 0.256 | 0.175 | 0.258 | 0.183 | 0.266 | 0.180 | 0.264 | 0.175 | 0.259 | 0.287 | 0.366 | 0.193 | 0.292 |
| | 192 | **0.232** | 0.293 | 0.235 | **0.292** | 0.237 | 0.299 | 0.236 | 0.299 | 0.237 | 0.299 | 0.247 | 0.305 | 0.250 | 0.309 | 0.241 | 0.302 | 0.414 | 0.492 | 0.284 | 0.362 |
| | 336 | **0.288** | **0.325** | 0.293 | 0.329 | 0.299 | 0.398 | 0.298 | 0.339 | 0.298 | 0.340 | 0.307 | 0.342 | 0.311 | 0.348 | 0.305 | 0.343 | 0.597 | 0.542 | 0.369 | 0.427 |
| | 720 | **0.385** | **0.390** | 0.395 | 0.391 | 0.397 | 0.394 | 0.395 | 0.397 | 0.391 | 0.396 | 0.407 | 0.399 | 0.412 | 0.407 | 0.402 | 0.400 | 1.730 | 1.042 | 0.554 | 0.522 |
| | AVG | **0.267** | **0.313** | 0.274 | 0.316 | 0.276 | 0.321 | 0.275 | 0.323 | 0.275 | 0.323 | 0.286 | 0.328 | 0.288 | 0.332 | 0.281 | 0.326 | 0.757 | 0.610 | 0.350 | 0.401 |
| Traffic | 96 | **0.392** | **0.252** | 0.496 | 0.331 | 0.448 | 0.309 | 0.455 | 0.291 | 0.462 | 0.285 | 0.651 | 0.391 | 0.395 | 0.268 | 0.462 | 0.295 | 0.522 | 0.290 | 0.650 | 0.396 |
| | 192 | 0.429 | **0.271** | 0.491 | 0.331 | 0.455 | 0.307 | 0.460 | 0.291 | 0.473 | 0.296 | 0.602 | 0.363 | **0.417** | 0.276 | 0.466 | 0.296 | 0.530 | 0.293 | 0.598 | 0.370 |
| | 336 | 0.450 | **0.275** | 0.505 | 0.334 | 0.472 | 0.313 | 0.473 | 0.302 | 0.490 | 0.296 | 0.609 | 0.366 | **0.433** | 0.283 | 0.482 | 0.304 | 0.558 | 0.305 | 0.605 | 0.373 |
| | 720 | 0.493 | **0.306** | 0.541 | 0.343 | 0.508 | 0.332 | 0.500 | 0.321 | 0.506 | 0.313 | 0.647 | 0.385 | **0.467** | 0.302 | 0.514 | 0.322 | 0.589 | 0.328 | 0.645 | 0.394 |
| | AVG | 0.441 | **0.276** | 0.508 | 0.335 | 0.471 | 0.315 | 0.472 | 0.301 | 0.484 | 0.297 | 0.627 | 0.376 | **0.428** | 0.282 | 0.481 | 0.304 | 0.550 | 0.304 | 0.625 | 0.383 |
| ECL | 96 | **0.137** | 0.241 | 0.168 | 0.251 | 0.151 | 0.245 | 0.156 | 0.252 | 0.153 | 0.247 | 0.200 | 0.278 | 0.148 | **0.240** | 0.181 | 0.270 | 0.219 | 0.314 | 0.197 | 0.282 |
| | 192 | **0.159** | **0.247** | 0.176 | 0.262 | 0.163 | 0.256 | 0.175 | 0.269 | 0.166 | 0.256 | 0.200 | 0.280 | 0.162 | 0.253 | 0.188 | 0.284 | 0.231 | 0.322 | 0.196 | 0.285 |
| | 336 | **0.174** | **0.266** | 0.195 | 0.278 | 0.180 | 0.274 | 0.190 | 0.284 | 0.185 | 0.277 | 0.214 | 0.295 | 0.178 | 0.269 | 0.204 | 0.293 | 0.246 | 0.337 | 0.209 | 0.301 |
| | 720 | **0.206** | **0.296** | 0.235 | 0.310 | 0.224 | 0.311 | 0.246 | 0.324 | 0.225 | 0.310 | 0.255 | 0.327 | 0.225 | 0.317 | 0.246 | 0.324 | 0.280 | 0.363 | 0.245 | 0.333 |
| | AVG | **0.169** | **0.263** | 0.194 | 0.275 | 0.180 | 0.271 | 0.192 | 0.282 | 0.182 | 0.272 | 0.217 | 0.295 | 0.178 | 0.270 | 0.205 | 0.290 | 0.244 | 0.334 | 0.212 | 0.300 |
| Weather | 96 | **0.157** | 0.204 | 0.161 | **0.198** | 0.162 | 0.207 | 0.159 | 0.206 | 0.163 | 0.209 | 0.166 | 0.213 | 0.174 | 0.214 | 0.177 | 0.218 | 0.158 | 0.230 | 0.196 | 0.255 |
| | 192 | 0.209 | 0.242 | 0.207 | **0.241** | 0.209 | 0.252 | **0.205** | 0.249 | 0.208 | 0.250 | 0.213 | 0.254 | 0.221 | 0.254 | 0.225 | 0.259 | 0.206 | 0.277 | 0.237 | 0.296 |
| | 336 | 0.259 | 0.289 | 0.261 | **0.281** | 0.263 | 0.292 | 0.263 | 0.291 | **0.251** | 0.287 | 0.269 | 0.294 | 0.278 | 0.296 | 0.278 | 0.297 | 0.272 | 0.335 | 0.283 | 0.335 |
| | 720 | 0.343 | 0.345 | 0.343 | **0.334** | 0.344 | **0.334** | 0.344 | 0.345 | **0.339** | 0.341 | 0.346 | 0.343 | 0.358 | 0.347 | 0.354 | 0.348 | 0.398 | 0.418 | 0.345 | 0.381 |
| | AVG | 0.242 | 0.270 | 0.243 | **0.264** | 0.244 | 0.274 | 0.243 | 0.273 | **0.240** | 0.271 | 0.249 | 0.276 | 0.258 | 0.278 | 0.259 | 0.348 | 0.259 | 0.315 | 0.265 | 0.317 |
| Exchange | 96 | 0.090 | 0.203 | 0.084 | 0.200 | **0.081** | **0.199** | 0.082 | 0.200 | 0.082 | **0.199** | 0.084 | 0.203 | 0.086 | 0.206 | 0.088 | 0.205 | 0.256 | 0.367 | 0.088 | 0.218 |
| | 192 | 0.175 | 0.307 | **0.171** | **0.293** | 0.171 | 0.294 | 0.175 | 0.297 | 0.177 | 0.297 | 0.177 | 0.298 | 0.177 | 0.299 | 0.176 | 0.299 | 0.470 | 0.509 | 0.176 | 0.315 |
| | 336 | 0.321 | 0.405 | 0.314 | 0.406 | 0.321 | 0.409 | 0.325 | 0.413 | 0.324 | 0.408 | 0.321 | 0.410 | 0.331 | 0.417 | **0.301** | **0.397** | 1.268 | 0.883 | 0.313 | 0.427 |
| | 720 | 0.838 | 0.689 | **0.776** | **0.656** | 0.837 | 0.688 | 0.840 | 0.690 | 0.837 | 0.691 | 0.828 | 0.685 | 0.847 | 0.691 | 0.901 | 0.714 | 1.767 | 1.068 | 0.839 | 0.695 |
| | AVG | 0.356 | 0.401 | **0.336** | **0.389** | 0.352 | 0.397 | 0.356 | 0.400 | 0.355 | 0.399 | 0.353 | 0.399 | 0.360 | 0.403 | 0.367 | 0.404 | 0.940 | 0.707 | 0.354 | 0.414 |

