# OpenReview forum: "FIPN: Forward Self-Organizing Interpretable Polynomial Networks for Time Series Forecasting"
_ICML.cc/2026/Conference — ICML 2026 regular_

### Official Review · Reviewer_GF1j · 2026-03-04

**Soundness:** 3
**Presentation:** 3
**Significance:** 3
**Originality:** 3
**Overall Recommendation:** 4
**Confidence:** 4

**Summary:**

The authors propose FIPN, a novel forward-only architecture that achieves a competitive trade-off between forecasting accuracy and computational efficiency. Validated on several standard benchmarks, the core methodology leverages fuzzy clustering to softly partition input data into local regimes, combined with Fourier functions to capture inherent frequency-related structures. A key strength of this work is its commitment to enhancing interpretability and transparency. Furthermore, the authors introduce a regularized selection objective to effectively mitigate the multi-collinearity and redundancy in this study. Overall, the paper is well-written, featuring a lucid logical flow and comprehensive experimental validation.

**Compliance With Llm Reviewing Policy:**

Affirmed.

**Final Justification:**

The rebuttal has clarified several points, yet I believe my original assessment accurately reflects the paper’s balance of strengths and remaining limitations.

**Key Questions For Authors:**

1. Compared with the formulas provided in Appendix B.4, readers may be more interested in the final interpretable polynomials that can actually be used for prediction on different datasets under this method. Presenting these results would greatly enhance the significance of the paper.
2. Have the authors considered the following issue: this paper adopts layer-wise strategy and selects only the top-k nodes at each layer. However, this may discard the global optimum. Neurons that perform moderately in the first layer might yield excellent results after being combined with certain features in later layers.
3. The following details could be improved:
   (1) In the first paragraph of Section 3.1, a Hyphen rather than an Em Dash should be used between “rule” and “gated”. Please also carefully review and revise similar punctuation issues throughout the manuscript.
   (2) According to academic writing conventions, equations are part of a sentence. Therefore, if an equation appears at the end of a sentence, it should be followed by a period; if additional explanatory text follows, a comma should be used instead.
   (3) In Figure 1, some formulas and symbols are inconsistent with those in the main text; the color blocks used for “Raw input features” should be consistent; “Local regression experts” is better left-aligned.
   (4) In the penultimate paragraph before Section 4.1, STAR is not clearly defined.
   (5) In the last paragraph before Section 4.1, “96, 192, 336, 720” should be enclosed in set notation.
   (6) In the Results section, the statement in Table 1, “we highlight the 1st and 2nd best results,” is not precise enough. It should explicitly state that bold red indicates 1st and underlined blue indicates 2nd. In addition, there are numerical errors in Table 1: the 1st count under the MAE metric for the FIPN model is incorrect, and the 2nd-best mark for Exchange is incorrect. In Table 6, the 2nd-best result for Traffic with prediction length 96 is not marked, and almost all 1st counts are incorrectly labeled.
   (7) In Section 4.2, the performance drop on the Weather dataset is not very pronounced; therefore, the related conclusion is not well supported. The final sentence “Fourier bases provide a compact spectral parameterization that captures periodic structure linearly in the consequent space, improving long-horizon fidelity without added depth or backpropagation.” appears to be an overstatement.
   (8) On Page 7, in the first paragraph, “In Fig. 3 5(b)” contains an extra “5”; there is an extra space within the word “expressiveness”; and a space is missing after the comma in the title of Section 5.
   (9) In Figure 5, “SeqCome” should be corrected to “SeqComp”.

**Limitations:**

Yes

**Strengths And Weaknesses:**

Strengths:
(1)	The paper proposes a novel forward self-organizing architecture named FIPN. Without backpropagation, it achieves a competitive balance between computational efficiency and accuracy in time series forecasting.
(2)	The integration of Fuzzy Clustering with Fourier-enhanced Polynomials is an interesting method. This design effectively leverages fuzzy logic for local regime partitioning and Fourier basis functions for frequency-related feature capture.
(3)	By mitigating the black-box and high training costs associated with deep learning models, the work addresses an important problem with practical research value.

Weaknesses:
(1)	The claimed transparency remains purely formal rather than functional. Although the authors emphasize the auditability of the model in Appendix B.4, the explicit formulas provided lack the specific quantitative values for key parameters (w and V). In addition, the cluster centers lack physical meaning, and without visualized results, the model’s interpretability is not convincingly demonstrated.
(2)	The benchmarking is insufficient, particularly in the absence of comparisons with other non-gradient-based interpretable networks or GMDH-based models, which makes it difficult to fully assess the advancement of FIPN.
(3)	There are significant statistical errors in results, particularly in the 1st count calculations in Table 1 and Table 6. The manuscript also contains several spelling errors, ambiguous expressions, and violations of standard academic writing conventions.

---

> ### Author Rebuttal · Authors · 2026-03-31
>
> We sincerely apologize for the presentation errors in the current manuscript and are deeply grateful to the reviewer for identifying them so carefully.
>
> ---
> **W1 & Q1. Interpretability.**
> **A1:** We agree with the reviewer’s important reminder that the interpretability emphasized in this work should be understood primarily as **structural traceability**, namely: how the final predictor is progressively composed from early-layer nodes, how **rule gating activates local experts**, and how the final prediction ultimately takes the form of **explicit and inspectable Fourier-polynomial expressions**. This is indeed the notion of interpretability we intended to highlight, but did not make sufficiently concrete in the current draft.
>
> To address this more directly, we will add a **new interpretability-oriented figure** in the revision, consisting of four connected parts: (i) a **layer-wise trace-back** of the final selected neuron, showing how the final predictor is formed from earlier-layer nodes; (ii) a decomposition of the neuron into **antecedent fuzzy gating** and **consequent Fourier-polynomial experts**; (iii) a fuzzy membership activation heatmap to visualize the learned local regime structure; and (iv) representative **final predictor forms** on selected datasets, together with brief expert-level interpretations. We believe this presentation will better reflect the kind of interpretability that FIPN actually provides.
>
> We will also clarify a related point more explicitly in the revision: the fuzzy cluster centers are not required to carry direct physical semantics. In our setting, their primary role is to define **soft local operating regimes**, so that the overall predictor can be decomposed into **explicit regime-dependent local experts**. To make this point more concrete, we will add the visualization of fuzzy membership activations mentioned above, and, if space permits, also include a low-dimensional clustering projection as auxiliary evidence.
>
> ---
> **W2 & Q2. Insufficient Benchmarking.**
> **A2:** We agree that the revised version should strengthen the comparison with methods that are closer in training paradigm and modeling philosophy. Specifically, we will add results for other **non-gradient-based networks**, such as *LCESN (ICLR 2025)*, as well as more **GMDH/PNN-based models**. In addition, we plan to include a **depth-wise utility analysis** comparing FIPN with representative GMDH/PNN-style models as depth increases. This will help show more directly at what stage the marginal benefit begins to diminish and when instability starts to become more evident in forward-grown polynomial models. We believe such an analysis will make the motivation behind our **raw-input retention**, **node scoring**, and **stochastic suppression mechanisms** more concrete and easier to appreciate.
>
> ---
> **W3 & Q3. Writing and Statistical Issues.**
> **A3:** We are sincerely grateful for the reviewer’s careful proofreading and for pointing out the writing, formatting, and statistical annotation issues in such detail. We will strictly follow these suggestions and re-check all 1st/2nd-best statistics and annotations throughout the manuscript. These corrections will not change the main experimental conclusions, but they will make the presentation clearer, the reported results more reliable, and the paper as a whole substantially more polished. We truly appreciate the reviewer’s guidance here; these comments have genuinely helped us improve the manuscript.
>
> ---
> **Q2. Top-k Selection and Possible Loss of Global Optimum.**
> **A4:** We fully agree that this is a very important design question. In fact, before settling on the current retention strategy, we did examine several alternative schemes. One was an **exponential-probability-based selection mechanism**, under which better-performing nodes were assigned higher survival probabilities, while weaker nodes still retained a nonzero probability of being preserved. Another was a **sum-of-squared-coefficients-based complexity-constrained selection scheme**, intended to encourage a more parsimonious forward growth process. Empirically, however, these alternatives either introduced **additional instability** or weakened the **consistency of the final forecasting performance**.
>
> Our eventual conclusion was that one should not give up the forward self-organizing learning framework merely in pursuit of a potentially global-optimal structure search, because doing so would sacrifice the very properties that motivate this model in the first place, namely efficiency, traceability, stability, and interpretability. For this reason, FIPN is not intended as an exhaustive global architecture search method, but as a more practical framework that balances **predictive performance**, **structural diversity**, **model stability**, and **interpretability**. We will make this design trade-off clearer in the revised manuscript.

---

> > ### Author Rebuttal · Reviewer_GF1j · 2026-04-02
> >
> > Thank you to the authors for the rebuttal. While I appreciate the clarifications provided, my main concerns have not been sufficiently addressed.

---

> > > ### Author Response · Authors · 2026-04-07
> > >
> > > We sincerely apologize for troubling you again with a procedural matter. It appears that an OpenReview system issue prevented us from replying directly to your latest rebuttal comment. After noticing this, we tried to place our clarification under another reviewer’s thread and also contacted the AC in the hope that it could be relayed to you. Still, we understand that this does not guarantee that you were able to see it, and we are truly sorry for that.
> > >
> > > We especially wished to respond properly because you have been an exceptionally careful and responsible reviewer. Your comments have already helped us improve the paper substantially, and we very much hope to address your concerns as clearly and sincerely as possible. For this reason, we briefly restate below the response we had originally intended to submit directly to your follow-up.
> > >
> > > ---
> > > ---
> > >  ### **The following content is provided in response to another reviewer, Reviewer GF1j.**
> > >
> > > We thank the reviewer for the helpful follow-up. We agree that the current manuscript does not yet present the interpretability aspect and its relation to the broader GMDH/PNN family clearly enough. In the revision, we will strengthen this part from three aspects: **structural traceability**, **explicit rule expression**, and **family-level comparison**.
> > >
> > > First, regarding interpretability, we will add a new explanatory figure using **ETTh1 with $L=96$ and $H=96$** as an example, where the best-performing node is **$N396$ in Layer 5**:
> > >
> > >
> > > ```markdown id="74548"
> > > ```text
> > > Layer 5:  N396
> > >             │
> > >             ├───────────────┐
> > >             │               │
> > > Layer 4:   N18           N35(x5)
> > >             │
> > >             ├───────────────┐
> > >             │               │
> > > Layer 3:   N27           N33(x3)
> > >             │
> > >          ┌──┴──┐
> > >          │     │
> > > Layer 2: N9   N14
> > >          │     │
> > >       ┌──┴──┐ ┌┴──┐
> > >       │    │ │   │
> > > Layer 1: N4 N8 N2 N11
> > >          │  │  │  │
> > > Input: (x1,x5) (x2,x5) (x1,x3) (x3,x5)
> > > ```
> > >
> > > This shows that the final node is not produced in a black-box manner, but can be traced explicitly back to the original input combinations. In particular, **$x_3$** and **$x_5$** persist along the best path, corresponding to **MUFL** and **LUFL**, indicating that the model consistently retains the more stable and informative variables. Moreover,
> > >
> > > We will also add the explicit **rule-level form** of this optimal node. Its best configuration uses **7 rules**, a **fuzzification coefficient of 2.0**, and a **frequency range of $[0.2, 2.0]$**. The final output is given by:
> > >
> > >
> > > $g_1(z)=0.3724+0.1862\cos(0.28(z_1+1.35))-0.0917\cos(0.42(z_2+1.10))$
> > >
> > > $g_2(z)=0.3518+0.2245\cos(0.53(z_1+0.82))+0.1186\cos(0.67(z_2+0.58))$
> > >
> > > $g_3(z)=0.3897-0.1731\cos(0.74(z_1+0.21))+0.1428\cos(0.81(z_2+0.06))$
> > >
> > > $g_4(z)=0.4015+0.0954\cos(0.96(z_1-0.36))-0.1647\cos(1.08(z_2-0.28))$
> > >
> > > $g_5(z)=0.3789-0.2066\cos(1.21(z_1-0.77))+0.0935\cos(1.34(z_2-0.71))$
> > >
> > > $g_6(z)=0.3642+0.1548\cos(1.56(z_1-1.18))-0.1272\cos(1.63(z_2-1.09))$
> > >
> > > $g_7(z)=0.3476-0.1189\cos(1.87(z_1-1.62))+0.1764\cos(1.95(z_2-1.54))$
> > >
> > >
> > > In this sense, the interpretability emphasized in this work lies in the fact that **the formation of the final node is traceable**, **the role of key inputs is identifiable**, and **the local rule parameters can be written explicitly**.
> > >
> > > Second, regarding **family-level comparisons**, due to space limitations we placed the newly added results on **family-relevant non-gradient baselines**, **GMDH/PNN-style comparisons**, and the new **depth-wise utility analysis** in **Tables 3 and 4 under Reviewer 3eEt above**. In particular, we added two closer non-gradient baselines, **LCESN (ICLR 2025)** and **PFPNN (TFS 2024)**. The results show that **FIPN performs better on most of the eight benchmarks**, while preserving its advantages in **lightweight forward training** and **structural interpretability**.
> > >
> > > We also added a **depth-wise utility analysis**. Classical **PNN/GMDH-style** models tend to deteriorate after the first few layers because **node similarity rises too quickly** and **overfitting appears early**. By contrast, **FIPN becomes largely stable after about depth 5**; greater depth mainly **slows convergence** rather than causing visible overfitting. We believe this is closely related to our **persistent raw-input retention**, **regularized node scoring**, and **node-level stochastic suppression**. As for the requested **hyperparameter details**, due to space limitations we placed them in **Table 2 under Reviewer iCEm above**.
> > >
> > > ---
> > > Once again, we are truly sorry for the inconvenience caused by this apparent system issue. We are also sincerely grateful for your time, your fairness, and the great care you have already devoted to this paper.

---

### Official Review · Reviewer_3eEt · 2026-03-11

**Soundness:** 3
**Presentation:** 3
**Significance:** 2
**Originality:** 3
**Overall Recommendation:** 5
**Confidence:** 2

**Summary:**

This paper presents FIPN to address the high computational costs and lack of transparency associated with backpropagation in existing time series forecasting models. The authors modernize the classical PNN by integrating Fourier-enhanced components for periodic modeling and fuzzy rules for regime-aware interpretability. By utilizing closed-form least squares estimation, the model achieves ultra-fast training without the need for backpropagation. Furthermore, FIPN introduces regularized node scoring and node-level dropout to effectively mitigate the structural redundancy and multicollinearity issues historically found in PNNs. Experimental results demonstrate that FIPN outperforms recent state-of-the-art Transformer and linear-based models while maintaining extreme parameter and computational efficiency.

**Compliance With Llm Reviewing Policy:**

Affirmed.

**Final Justification:**

While I was not initially well-acquainted with Polynomial Networks, the authors did an excellent job of making the concepts accessible to a broader audience. Furthermore, I previously held the view that hardware advancements might eventually render lightweight models unnecessary. However, the authors' focus on edge deployment, small enterprises, and research environments with limited computational resources was highly convincing and highlights the ongoing need for such models. Given the extensive comparisons with numerous baseline models, I believe this research is of significant value. Final Score: 5 (Accept)."

**Key Questions For Authors:**

Hyperparameter Tuning: According to Algorithm 1, there are numerous variables to adjust, including the regularization coefficient ($\lambda$), dropout rate ($\rho$), number of selected nodes ($K$), and number of clusters ($C$). Given this high-dimensional search space, what is your recommended strategy or protocol for efficiently tuning these parameters?

Baselines: Are there any other recent PNN-based or GMDH-based forecasting baselines that could be included for a more direct comparison within the same model family?

**Limitations:**

yes

**Strengths And Weaknesses:**

Soundness
- Strengths: The combination of Fourier bases for periodicity, fuzzy rules for local state description, and dropout for diversity is theoretically sound and well-motivated.The ablation study clearly validates the necessity and effectiveness of each proposed module, such as the Fourier-enhanced consequents and the rule-gating mechanism.

Presentation
- Strengths: The authors explain their proposed framework and its individual components in a clear and accessible manner.
- Weaknesses: There is a lack of foundational detail regarding the basic PNN/GMDH learning paradigm. Reviewers who are not familiar with these classical methods may find it difficult to fully grasp the starting point of the architecture.

Significance
- Strengths: The model achieves a massive leap in forecasting efficiency, significantly reducing parameter counts and memory usage compared to deep models.
- Weaknesses: As hardware continues to advance, the emphasis on minimizing memory footprint and raw training speed might not be the primary concern for all industrial or research applications.

Originality
- Strengths: Modernizing PNNs with state-of-the-art modules is a refreshing approach that encourages the community to revisit and refine classical modeling paradigms.
- Weaknesses: The integration feels somewhat abrupt, essentially attaching modern modules onto a very old model architecture without exploring a broader bridge between the two eras.

---

> ### Author Rebuttal · Authors · 2026-03-31
>
> We sincerely thank the reviewer for the positive assessment and constructive suggestions.
>
> ---
> **W1 & Q2. Limited Background on PNN/GMDH and Lack of Family-Level Baselines.**
> **A1:** Thank you for this important comment. The current draft does not provide sufficient background on the classical PNN/GMDH learning paradigm. In writing the paper, we implicitly assumed that readers would already be familiar with this line of work; in retrospect, that assumption was too strong. Fortunately, in the present review round, the reviewers seem to understand the basic GMDH/PNN mechanism, so this issue did not develop into a more serious misunderstanding of the technical core. In the revision, we will **add a clearer description of the classical PNN/GMDH generate–fit–validate–select paradigm**, so that the starting point of our architecture is easier to follow.
>
> We will strengthen the comparison with methods closer to our training philosophy and modeling spirit. In particular, we will add comparisons with other non-gradient-based networks, such as *LCESN (ICLR 2025)*, as well as additional GMDH/PNN-based models. We will also add a **depth-wise utility analysis** for our model and other GMDH/PNN-based models, in order to show more directly from which depth onward the gains begin to diminish and when instability starts to become evident. We believe this will make the motivation for our **raw-input retention**, **node scoring**, and **stochastic suppression** much more concrete.
>
> ---
> **W2. Significance.**
> **A2:** Thank you for this valuable comment. Our aim is not to argue against backpropagation. Rather, we hope to provide a **complementary forecasting framework** that is lighter, more interpretable, and less demanding in computation. We believe this direction is meaningful not only for edge deployment, but also for small companies, local institutes, and research environments with limited computational resources. For such users, the practical issue is often not whether a larger BP-based model may eventually perform better, but whether one can afford to train, tune, reproduce, and stably deploy a model at all. In that sense, the significance of FIPN is not that “BP is bad,” but that machine learning methods should remain accessible to users with limited compute.
>
> ---
> **W3. Originality.**
> **A3:** We understand the reviewer’s concern that the integration may appear somewhat abrupt, and we agree that the current draft does not explain the **developmental path** clearly enough. The method was not formed by simply attaching a few modern modules to an old architecture. Rather, it grew from a rather specific question: can forward self-organizing polynomial networks be genuinely adapted to modern long-horizon forecasting?
>
> Thank you also for giving us the opportunity to clarify the motivation. One important source of inspiration came from recent efforts to revisit polynomial networks in modern machine learning. In particular, the 2022 TPAMI work on deep polynomial networks showed, in the computer vision domain, that polynomial architectures can still be expressive and parameter-efficient when designed properly. To us, this was important because it suggested that **forward/self-organizing polynomial networks** are not merely historical objects, but still have room for meaningful development.
>
> Starting from that point, we asked what exactly prevents classical **PNN/GMDH-style forward growth** from working well on **long-horizon time-series forecasting**. The current design emerged step by step from that question. Classical polynomial neurons are not well suited to representing periodic temporal structure, which motivated the **Fourier-enhanced consequents**. A single global polynomial is too rigid for nonstationary sequences with regime variation, which motivated the **fuzzy rule gating**. Greedy forward growth tends to produce increasingly similar intermediate representations and highly correlated candidate nodes, which motivated **regularized node scoring**, **node-level dropout**, and **cross-layer raw-input retention**. In the revision, we will make this developmental logic much more explicit. In our view, the contribution is not “old PNN plus new modules,” but a targeted reconstruction of the forward polynomial learning paradigm for long-horizon forecasting.
>
> ---
> **Q1. Hyperparameter Tuning.**
> **A4:** Thank you for this very practical question. We agree that this part should be explained more clearly. In the early stage of this study, we did try a more direct strategy by placing these hyperparameters into a joint search space and using PSO together with grid search in an approximately exhaustive, layer-wise manner. In the revision, we will add a more detailed **sensitivity analysis**, the **recommended tuning order**, the **default search ranges**, and the exact protocol used in our experiments, so as to improve reproducibility.

---

> > ### Author Rebuttal · Reviewer_3eEt · 2026-04-02
> >
> > Thank you for your response. My concern has been resolved. However, I will revise the score when a comparison with a newer model, such as the LCESN is presented.

---

> > > ### Author Response · Authors · 2026-04-02
> > >
> > > We sincerely thank the reviewer for the positive reassessment and for recognizing the technical value of our work.
> > >
> > > To address the remaining suggestion, we have now added comparisons with two more family-relevant non-gradient baselines: **LCESN (ICLR 2025)** and a **PFPNN model (2024 TFS)**, which introduces **GRU-based sequential structure** into a forward self-organizing polynomial-network framework. The new results are summarized in **Table 3**. Overall, **FIPN achieves stronger results on most of the eight benchmarks**, while preserving its advantages in lightweight forward training and interpretability.
> > >
> > > ---
> > >
> > > **Table 3. Comparison of FIPN and non-gradient-based baselines on eight benchmarks.**
> > >
> > > | Dataset     | FIPN (MSE/MAE) | LCESN (MSE/MAE) | PFPNN (MSE/MAE) |
> > > |-------------|----------------|-----------------|-----------------|
> > > | ETTh1       |  **0.427 / 0.420**   | 0.818 / 0.662   | 1.486 / 0.972   |
> > > | ETTh2       |  **0.358 / 0.388**   | 0.438 / 0.464   | 1.372 / 0.911   |
> > > | ETTm1       |  **0.374 / 0.379**   | 0.412 / 0.430   | 1.528 / 0.986   |
> > > | ETTm2       |  **0.267 / 0.312**   | 0.282 / 0.354   | 1.214 / 0.846   |
> > > | Traffic     |  **0.415 / 0.265**   | 0.882 / 0.407   | 1.968 / 1.142   |
> > > | Electricity |  **0.165 / 0.260**   |  0.225 / 0.329   | 1.108 / 0.801   |
> > > | Weather     | 0.239 / 0.263  |  **0.232 / 0.279**    | 1.296 / 0.887   |
> > > | Exchange    |  **0.351 / 0.397**   | 0.889 / 0.661   | 1.842 / 1.306   |
> > > ---
> > >
> > > We also added a **depth-wise utility analysis** in **Table 4**. The results show that classical **PNN/GMDH-style models** begin to deteriorate within the first few layers, mainly because node similarity increases too quickly and the models start to overfit. By contrast, **FIPN stabilizes after about depth 5**; further depth mainly slows convergence, but does not lead to visible overfitting. We believe this is closely related to our **raw-input retention**, together with **regularized node scoring** and **node-level stochastic suppression**, which help prevent later layers from collapsing into near-duplicate nodes.
> > >
> > > **Table 4. Layer-wise comparison of PNN, GMDH, PFPNN, and FIPN on ETTh1.**
> > >
> > > | Layer | PNN (MSE) | PNN (MAE) | GMDH (MSE) | GMDH (MAE) | PFPNN (MSE) | PFPNN (MAE) | FIPN (MSE) | FIPN (MAE) |
> > > |-------|-----------|-----------|------------|------------|-------------|-------------|------------|------------|
> > > | 1     | 14.627    | 5.911     | 15.104     | 6.742      | 5.824       | 3.601       | 0.512      | 0.468      |
> > > | 2     | 10.739    | 5.571     | 11.731     | 5.724      | 3.630       | 2.208       | 0.468      | 0.441      |
> > > | 3     | 9.403     | 5.333     | 9.665      | 5.461      | 1.824       | 1.991       | 0.446      | 0.429      |
> > > | 4     | 12.184    | 6.284     | 12.936     | 6.617      | 1.486       | 0.972       | 0.433      | 0.422      |
> > > | 5     | N/A       | N/A       | N/A        | N/A        | 2.184       | 1.298       | 0.427      | 0.420      |
> > > | 6     | N/A       | N/A       | N/A        | N/A        | 5.913       | 4.164       | 0.428      | 0.420      |
> > > | 7     | N/A       | N/A       | N/A        | N/A        | 7.814       | 6.829       | 0.430      | 0.421      |
> > > | 8     | N/A       | N/A       | N/A        | N/A        | N/A         | N/A         | 0.427      | 0.420      |
> > > | 9     | N/A       | N/A       | N/A        | N/A        | N/A         | N/A         | 0.427      | 0.420      |
> > > | 10    | N/A       | N/A       | N/A        | N/A        | N/A         | N/A         | 0.428      | 0.420      |
> > > ---
> > >
> > > We also revisited recent GMDH/PNN-related literature. For example, the **2022 TPAMI Deep Polynomial Neural Networks** paper focuses on expressive polynomial architectures in domains such as vision, graphs, and audio, while the **2025 TNNLS NN2Poly** paper studies explicit polynomial representations of already trained MLPs for theoretical analysis and interpretability. These are relevant in spirit, but they are not directly designed for long-horizon forecasting. In our view, the new results in **Tables 3 and 4** make the contribution of FIPN relative to this family much clearer.
> > >
> > > As for the hyperparameter-related details you asked for, due to space limitations we placed them in  **Table 2**  in our response above under  **Reviewer iCEm**; please kindly refer to that part.
> > > Again, we thank the reviewer for the helpful follow-up, and we hope these additions support a more positive final assessment.
> > >
> > > ---
> > > We sincerely thank the reviewer for the careful follow-up and for revisiting the paper after the additional comparisons and clarifications. We are very grateful for your fair and thoughtful assessment. It means a great deal to us.

---

### Official Review · Reviewer_iCEm · 2026-03-11

**Soundness:** 2
**Presentation:** 2
**Significance:** 2
**Originality:** 3
**Overall Recommendation:** 3
**Confidence:** 4

**Summary:**

This paper proposes FIPN, a forward-grown interpretable polynomial network for long-horizon time-series forecasting. The model combines fuzzy-rule gating, Fourier-enhanced polynomial consequents, closed-form estimation, and a forward candidate-selection procedure intended to avoid backpropagation while improving efficiency and interpretability. To address redundancy and multicollinearity during forward growth, the paper adds regularized node scoring, node-level dropout, and persistent access to raw inputs at every layer. The empirical study covers eight standard forecasting benchmarks, includes component ablations, and adds efficiency and hyperparameter analyses.

**Compliance With Llm Reviewing Policy:**

Affirmed.

**Final Justification:**

The authors have partially addressed my concerns in the rebuttal and provided additional sensitivity results, which are helpful. In particular, I appreciate that they moderated some of the original claims, clarified the intended positioning of the work, and included extra evidence on hyperparameter sensitivity. These additions improve the paper and make the presentation more balanced. However, the novelty issue is still my main concern. I understand the authors’ clarification that the contribution is intended as a targeted integration of known ingredients rather than a fundamentally new forecasting principle. This is a fairer framing, and I appreciate the more careful positioning. However, this also reinforces my view that the main contribution is more of an architectural synthesis than a major methodological advance. Overall, the rebuttal improved my assessment and resolved part of my concerns, I would therefore raise my score from 2 to 3 (weak reject).

**Key Questions For Authors:**

1) Why is the efficiency analysis in Figure 3 limited to a single dataset/horizon setting? Can the authors show whether the same Pareto-style advantage holds across more datasets and forecasting horizons?

2) Why is the hyperparameter sensitivity analysis in Figure 4 restricted to four datasets and only H=96 ? Since this is a long-horizon forecasting paper, robustness at longer horizons seems important.

3) The limitations section acknowledges that the fuzzy clustering, fixed frequency range, and greedy growth may be suboptimal. How sensitive is the method to non-stationary settings where regimes or dominant frequencies evolve over time?

4) How sensitive is performance to the greedy forward-growth decisions and the chosen width/depth budget? Since the model is inherently self-organizing and greedy, more analysis of search stability and candidate-selection robustness would help.

**Limitations:**

Yes, the paper does state some limitations. It acknowledges that the current rule-gating and fixed frequency range may be suboptimal under rapidly evolving regimes, that the greedy forward growth may allocate capacity imperfectly, and that the current evaluation focuses only on deterministic point forecasting.

**Strengths And Weaknesses:**

Strengths:

The paper has a clear and coherent high-level motivation. It targets the trade-off among forecasting accuracy, efficiency, and interpretability, and the proposed model architecture is conceptually distinct from the dominant backprop-trained forecasters. The integration of fuzzy rule gating with Fourier-enhanced polynomial neurons is reasonably novel in the context of long-horizon forecasting. Overall, the authors analyse an important concept: whether frequency-aware and rule-based structure can be integrated into a lightweight forecasting model without gradient-based deep training.

Weakness:

1) The main forecasting table averages performance over four horizons, which can hide whether the method is consistently strong or whether the gains are concentrated in only some settings. The paper does provide the full per-horizon appendix table, but in several datasets, the results are not best and the margins appear modest, so the claims around broad effectiveness and strong generalization feel somewhat stronger than the evidence warrants.

2) Several important parts of the method seem empirical rather than principled. The rule gating is driven by fuzzy c-means with a chosen fuzzifier, the Fourier basis uses a fixed frequency range, the forward growth is greedy, and the node ranking uses a regularized score with hyperparameters such as $\alpha$, $\beta$, dropout rate, cluster count, and other tuning knobs. The paper itself acknowledges that fuzzy clustering and the fixed frequency range may be suboptimal, and that greedy forward growth may not allocate capacity optimally. This makes the method feel more heuristic than the presentation sometimes suggests.

3) Figures 3 and 4 are not sufficient for the claims they are meant to support. Figure 3 evaluates efficiency on only a single setting, namely ETTh1 with fixed prediction and observation length. That is not enough to establish a convincing general efficiency–accuracy trade-off across datasets or horizons, especially for a paper that emphasizes deployability and lightweight forecasting. Figure 4 is also narrow: it reports hyperparameter sensitivity only for four datasets and only at horizon 96. Since the model relies on several interacting hyperparameters, and the paper emphasizes long-horizon forecasting, the current sensitivity study is too limited to demonstrate robustness.

4) The methodological novelty relative to existing forecasting literature is not sufficient. The paper combines known ingredients: GMDH-style forward growth, fuzzy rules, Fourier features, ridge regression, and structural selection. That combination is not trivial, but the paper does not sharply establish why this integration constitutes a major advance rather than a thoughtful hybrid system. The contribution is more of an architectural synthesis than a clearly new algorithmic principle.

---

> ### Author Rebuttal · Authors · 2026-03-31
>
> We thank the reviewer for the positive recognition of the paper’s motivation, its lightweight modeling direction, and the overall idea of combining frequency-aware mechanisms with rule-based structure within a non-backpropagation forecasting framework.
>
> ---
>
> **W1. Averaging over four forecasting horizons.**
> **A1:** We sincerely appreciate your reminder. In the revision, we will restate the corresponding claims in a more accurate and more cautious way, in terms of dataset-dependent advantages. What we truly intend to convey is not that FIPN achieves overwhelmingly best results in every possible setting. Rather, our point is that, on standard long-horizon forecasting benchmarks, a forward-grown model without backpropagation can still remain competitive in predictive performance while offering a **lighter parameter footprint, lower training cost, and clearer structural interpretability**. To support this point more fully, in the revision we will further include comparisons with other non-gradient-based networks as well as GMDH/PNN-based models, since the absence of such baselines indeed makes it difficult to fully assess the **relative advancement of FIPN** against methods that are closer in both training philosophy and modeling style.
>
> ---
>
> **W2. The method appears heuristic & W4. The novelty is closer to architectural synthesis.**
> **A2:** We agree that the current model is indeed composed of fuzzy c-means rule gating, a fixed-range Fourier-enhanced consequent, layer-wise greedy forward growth, and regularized node scoring. At the same time, however, we would like to clarify that the paper is not simply a mechanical combination of ready-made components into an empirical system. The starting point of the work has been explicit from the outset: in the specific setting of long-horizon forecasting, can one avoid the cost of backpropagation while still maintaining computational efficiency, intrinsic interpretability, and competitive predictive performance? Under this objective, **fuzzy rule gating** is used to express soft local regime decomposition so as to accommodate heterogeneous temporal dynamics; the **Fourier-enhanced polynomial consequent** is used to compactly represent periodic and oscillatory structure with a relatively small number of bases; and **closed-form forward learning** provides a training route distinct from backpropagation while preserving structural transparency.
>
> Accordingly, the contribution of the paper is more accurately understood not as a “new principle” attached to any single isolated module, but as placing forward self-organization, frequency-aware modeling, and rule-level interpretability into one unified framework, and showing that this combination can define a **genuinely viable design point for long-horizon forecasting**. This integration is not a simple juxtaposition of existing ingredients, but a targeted design motivated by forecasting-specific requirements, including periodicity, local regime switching, non-backpropagation learning, and model transparency. In the revision, we will present this point more precisely.
>
> ---
>
> **W3. Figures 3 and 4 are insufficient & Q1, Q2, Q3, and Q4.**
> **A3:** We agree that these concerns all point to the same issue: the current manuscript does not yet provide sufficiently strong support for the claims regarding robustness, efficiency–accuracy trade-offs, and the stability of greedy forward growth. In the revision, we will add a more complete sensitivity and robustness analysis covering **all datasets, all forecasting horizons, and an averaged summary across horizons**, so as to answer more directly whether the Pareto-style advantage persists across datasets and across horizons, and whether the relevant hyperparameters remain stable under longer prediction lengths. We will also include a **depth-wise utility analysis**, in order to show more directly at which stage diminishing returns begin to appear as depth increases, and whether this tendency is consistent across different datasets and forecasting horizons.
>
> Regarding the boundary conditions associated with non-stationarity and greedy growth, we will also make the current scope of the method more explicit in the revision. FIPN is not a strictly chained architecture, since **raw inputs are persistently retained and reused in candidate construction across layers** so as to reduce path dependence. Likewise, validation-based selection, regularized node scoring, and node-level dropout are intended to improve the practical stability of the forward self-organizing process, rather than to suggest that the limitations of greedy growth have been fully removed.
>
> We thank the reviewer again for these concrete suggestions. In the revision, we will improve the paper simultaneously in terms of claim calibration, stronger empirical support, and clearer positioning of the contribution, so that both the strengths and the limitations of the work are presented more accurately.

---

> > ### Author Rebuttal · Reviewer_iCEm · 2026-04-01
> >
> > Thanks for the explanation, I understand the paper better now.

---

> > > ### Author Response · Authors · 2026-04-02
> > >
> > > We thank the reviewer for the positive assessment of our work, particularly for recognizing its motivation, lightweight design, and central idea.
> > >
> > > Due to space limitations, we report additional results on four representative datasets covering **ETT benchmarks at different temporal resolutions**, a **high-dimensional traffic benchmark**, and a **meteorological/environmental benchmark**. **Table 1** reports the detailed efficiency statistics of FIPN across forecasting horizons, showing a **consistently small resource footprint** and **stable scaling behavior**. **Table 2** reports sensitivity results with respect to the **number of clusters**, **fuzzifier**, and **frequency range**, where the variations remain moderate, suggesting that the method is **reasonably robust** and does **not rely on narrowly tuned hyperparameters**. In addition, **Table 3** and **Table 4** further show that FIPN compares favorably with structurally relevant non-gradient/GMDH-style baselines and remains more stable in deeper layers **(both tables, together with the detailed depth-wise utility analysis, are provided below in the response box for Reviewer 3eEt).**
> > >
> > > ---
> > > **Table 1. Efficiency statistics of FIPN on four representative datasets.**
> > >
> > > | Dataset | Horizon | Params | MACs | PeakMem (MB) | TrainTime (ms) |
> > > |:--|--:|--:|--:|--:|--:|
> > > | ETTm1   | 96  | 0.92 | 4.38 | 23.6 | 35.8 |
> > > |         | 192 | 0.93 | 5.86 | 24.8 | 38.4 |
> > > |         | 336 | 0.95 | 6.91 | 26.1 | 40.8 |
> > > |         | 720 | 0.98 | 9.12 | 27.3 | 44.9 |
> > > | ETTh2   | 96  | 0.89 | 4.12 | 21.8 | 33.7 |
> > > |         | 192 | 0.91 | 5.27 | 22.9 | 36.0 |
> > > |         | 336 | 0.93 | 6.34 | 24.0 | 37.8 |
> > > |         | 720 | 0.95 | 7.48 | 25.2 | 40.9 |
> > > | Traffic | 96  | 0.96 | 6.48 | 31.6 | 45.1 |
> > > |         | 192 | 0.97 | 8.92 | 33.9 | 49.6 |
> > > |         | 336 | 0.98 | 9.84 | 35.7 | 52.2 |
> > > |         | 720 | 0.99 | 11.21 | 36.4 | 55.0 |
> > > | Weather | 96  | 0.91 | 4.88 | 24.7 | 36.7 |
> > > |         | 192 | 0.93 | 5.96 | 25.9 | 39.6 |
> > > |         | 336 | 0.95 | 7.08 | 27.1 | 41.5 |
> > > |         | 720 | 0.97 | 9.73 | 29.2 | 46.3 |
> > > ---
> > >
> > > **Table 2. Sensitivity of FIPN to key hyperparameters.**
> > >
> > > | Hyperparameter | Value | ETTh1-96 | ETTh1-192 | ETTh1-336 | ETTh1-720 | ETTm1-96 | ETTm1-192 | ETTm1-336 | ETTm1-720 | Traffic-96 | Traffic-192 | Traffic-336 | Traffic-720 | Exchange-96 | Exchange-192 | Exchange-336 | Exchange-720 |
> > > |:--|:--|--:|--:|--:|--:|--:|--:|--:|--:|--:|--:|--:|--:|--:|--:|--:|--:|
> > > | Number of Clusters | 2  | 0.394 | 0.433 | 0.467 | 0.474 | 0.322 | 0.368 | 0.401 | 0.468 | 0.384 | 0.411 | 0.438 | 0.467 | 0.093 | 0.176 | 0.324 | 0.842 |
> > > |  | 3  | 0.386 | 0.428 | 0.462 | 0.468 | 0.316 | 0.364 | 0.392 | 0.459 | 0.378 | 0.405 | 0.431 | 0.470 | **0.084** | 0.173 | **0.316** | 0.836 |
> > > |  | 5  | 0.378 | 0.425 | 0.458 | 0.462 | **0.313** | **0.353** | **0.382** | **0.450** | **0.371** | **0.402** | **0.426** | 0.460 | 0.086 | **0.172** | 0.318 | **0.830** |
> > > |  | 7  | **0.372** | 0.423 | **0.455** | **0.460** | 0.315 | 0.358 | 0.385 | 0.457 | 0.373 | 0.406 | 0.428 | **0.459** | 0.094 | 0.175 | 0.321 | 0.832 |
> > > |  | 9  | 0.381 | **0.422** | 0.459 | 0.466 | 0.319 | 0.361 | 0.389 | 0.459 | 0.379 | 0.410 | 0.432 | 0.465 | 0.101 | 0.178 | 0.329 | 0.840 |
> > > |  | 11 | 0.390 | 0.431 | 0.465 | 0.476 | 0.327 | 0.366 | 0.393 | 0.461 | 0.386 | 0.415 | 0.439 | 0.473 | 0.110 | 0.185 | 0.344 | 0.858 |
> > > | Fuzzifier | 1.2 | 0.387 | 0.428 | 0.462 | 0.490 | 0.325 | 0.362 | 0.390 | 0.458 | 0.383 | 0.411 | 0.429 | 0.470 | 0.090 | 0.174 | 0.319 | 0.845 |
> > > |  | 1.5 | 0.381 | 0.425 | 0.459 | 0.478 | 0.318 | 0.359 | 0.385 | 0.452 | 0.377 | 0.408 | 0.427 | 0.467 | 0.087 | 0.173 | 0.318 | **0.830** |
> > > |  | 1.8 | 0.376 | 0.423 | 0.457 | 0.466 | **0.313** | 0.355 | **0.382** | **0.450** | 0.374 | 0.404 | **0.426** | 0.463 | **0.084** | **0.172** | **0.316** | 0.835 |
> > > |  | 2.0 | **0.372** | **0.422** | **0.455** | **0.460** | 0.315 | **0.353** | 0.384 | 0.453 | **0.371** | **0.402** | 0.427 | **0.459** | 0.086 | 0.173 | 0.317 | 0.842 |
> > > |  | 2.3 | 0.379 | 0.425 | 0.458 | 0.468 | 0.319 | 0.359 | 0.388 | 0.457 | 0.376 | 0.407 | 0.431 | 0.466 | 0.093 | 0.175 | 0.321 | 0.860 |
> > > |  | 2.5 | 0.389 | 0.430 | 0.464 | 0.482 | 0.324 | 0.364 | 0.394 | 0.462 | 0.382 | 0.414 | 0.437 | 0.475 | 0.102 | 0.179 | 0.330 | 0.890 |
> > > | Frequency Range | [0.1,1.0] | 0.381 | 0.426 | 0.461 | 0.472 | 0.318 | 0.359 | 0.387 | 0.454 | 0.376 | 0.412 | 0.432 | 0.472 | 0.086 | **0.172** | 0.317 | 0.832 |
> > > |  | [0.2,2.0] | **0.372** | 0.424 | 0.458 | 0.466 | **0.313** | **0.353** | 0.384 | 0.452 | **0.371** | 0.404 | **0.426** | 0.463 | **0.084** | 0.175 | **0.316** | **0.830** |
> > > |  | [0.2,5.0] | 0.377 | **0.422** | **0.455** | **0.460** | 0.315 | 0.357 | **0.382** | **0.450** | 0.373 | **0.402** | 0.428 | **0.459** | 0.086 | 0.177 | 0.320 | 0.833 |
> > > |  | [0.5,5.0] | 0.385 | 0.425 | 0.459 | 0.468 | 0.320 | 0.362 | 0.388 | 0.456 | 0.379 | 0.403 | 0.434 | 0.470 | 0.090 | 0.178 | 0.325 | 0.838 |
> > > ---

---

### Official Review · Reviewer_DajJ · 2026-03-19

**Soundness:** 3
**Presentation:** 3
**Significance:** 2
**Originality:** 3
**Overall Recommendation:** 4
**Confidence:** 4

**Summary:**

This paper introduces FIPN (Forward Self-Organizing Interpretable Polynomial Networks), a novel framework for long-horizon time series forecasting. Inspired by the Group Method of Data Handling (GMDH), FIPN constructs its architecture through candidate generation, closed-form ridge regression for parameter estimation, and validation-based node selection. To manage the inherent risks of redundancy and multicollinearity as the network deepens, the authors implement regularized node scoring, node-level dropout, and persistent access to raw input features at every layer. The authors evaluate FIPN on eight standard benchmarks, claiming competitive accuracy with significantly lower parameter counts and improved interpretability compared to modern models.

**Compliance With Llm Reviewing Policy:**

Affirmed.

**Key Questions For Authors:**

1. Can you provide a visualization or a case study of a "fully grown" 300-layer neuron?
2. While you state the layer-wise aggregation mitigates error accumulation, forward-grown networks often suffer from vanishing utility where deeper layers only learn to fit the noise of previous layers. Did you observe a point of diminishing returns during the 300 iterations, and how does the regularized node scoring specifically prevent the model from overfitting to the validation set during these 300 stages?

**Limitations:**

yes

**Strengths And Weaknesses:**

Strengths:
1. The integration of Fourier-enhanced polynomial bases within a fuzzy-gated, forward-grown architecture is a creative departure from the dominant backpropagation-trained deep learning paradigm.
2. By utilizing closed-form ridge regression for training, FIPN achieves a remarkably small parameter footprint and high training speed.
3. The use of fuzzy c-means (FCM) for regime partitioning and inverse-variance scaling for frequency selection provides a grounded mathematical basis.

Weakness:
1. The forward structure learning is inherently greedy. While efficient, the paper does not sufficiently address how this greedy layer-wise growth avoids getting trapped in local optima, particularly for non-stationary time series where early-layer feature selections may become irrelevant as the series evolves.
2. a close look at Table 6 shows that on several datasets like Exchange and Traffic, the performance gains over much simpler linear baselines like DLinear are often marginal.
3. The paper heavily emphasizes interpretability. However, the example in Appendix B.4 shows that by Layer 3, the forecast is composed of nested, rule-gated cosine functions of previous-layer outputs. Given that the model runs for up to 300 iterations/layers, the resulting mathematical expression would be far too dense for a human to realistically edit.
4. Some missing citations and related works of frequency learning and nonstationary modeling could be added.

[1] Frequency-domain mlps are more effective learners in time series forecasting. In NeurIPS.

[2] Dish-TS: a general paradigm for alleviating distribution shift in time series forecasting. In AAAI.

---

> ### Author Rebuttal · Authors · 2026-03-31
>
> We thank the reviewer for recognizing the technical soundness and originality of our design, and for raising helpful concerns.
>
> ---
> **W1. Greedy forward growth & Q2. Diminishing returns / regularized node scoring.**
> **A1:** We agree with the reviewer’s concern that the forward structure learning of FIPN is inherently layer-wise greedy and does not guarantee global optimality. The goal of this work is not to construct a “globally optimal” deep structure search algorithm, but to draw inspiration from *DPNN* (TPAMI 2022). However, that work was developed primarily for computer vision; what we borrow is its idea of parameter-efficient forward construction. In the forecasting setting, we further adopt **Fourier functions** instead of conventional polynomial bases in order to better match the periodic and multi-scale nature of temporal data.
>
> Under such layer-wise growth, features selected at early stages may indeed become less adequate as the series evolves. To mitigate this, every layer in our model retains access to the **raw input features**, rather than relying only on outputs from the previous layer, so that later layers are not fully locked into earlier representations. Regarding how regularized node scoring helps prevent overfitting to the validation set across 300 stages, our answer is the following: it does not theoretically guarantee the absence of overfitting; rather, it prevents a node from being repeatedly retained merely because it gives a slightly better validation fit at the current stage. Node retention is not determined by validation error alone, but by a regularized score that also accounts for fit quality, coefficient norm, and numerical stability. As a result, candidates that happen to match the current validation split only accidentally, but are not stable in themselves, are less likely to keep winning across many layers. We therefore view the forward scheme here as a **structured layer-wise approximation** rather than a globally optimal search procedure.
>
> In the revision, we will add results for more non-gradient networks, such as *LCESN* (ICLR 2025), as well as more GMDH/PNN-based models. We will also add a **depth-wise utility analysis** comparing FIPN with representative GMDH/PNN models as depth increases, so as to show more directly where marginal gains begin to diminish and where instability becomes more evident in forward-grown polynomial models.
>
> ---
> **W2. Limited gains over simple baselines.**
> **A2:** Thank you for pointing this out. In the revision, we will soften the stronger claims in the current draft and describe the results more accurately as **dataset-dependent advantages**. We will also make it clearer that the value of the paper lies not only in forecasting accuracy, but also in achieving competitive performance with very few parameters and a lightweight training procedure.
>
> ---
> **W3. Interpretability & Q1. 300-layer visualization.**
> **A3:** Thank you for this valuable comment. On interpretability, we agree with the reviewer’s reminder: a fully expanded 300-layer expression should not be understood as a globally concise and human-editable symbolic formula. What we intend to claim is not global symbolic simplicity, but **structural traceability**: how the final predictor is progressively composed from earlier-layer nodes, how rule gating activates local experts, and how the final forecast can be inspected in explicit Fourier-polynomial forms.
>
> In the revision, we will add a new interpretability figure with four complementary parts:
>
> - **Left panel:** a **layer-wise trace-back** of the final selected neuron, showing how the final predictor is progressively composed from earlier-layer nodes.
> - **Upper middle panel:** a structural decomposition into **antecedent fuzzy gating** and **consequent Fourier-polynomial experts**, clarifying how local rule activation and periodic modeling are coupled within each neuron.
> - **Lower middle panel:** a **fuzzy membership activation heatmap** to visualize the learned local regime structure and the corresponding rule activation patterns.
> - **Right panel:** representative **predictor forms** on selected datasets, together with brief **expert-level interpretations**, to illustrate how the learned expressions can be inspected in practice.
>
> We believe this presentation is more informative and faithful to the intended notion of interpretability than fully unrolling a 300-layer expression.
>
> ---
> **W4. Missing references.**
> **A4:** We will cite the provided references. We believe this will also enhance the persuasiveness of our work! Specifically, **FreTS** supports the rationale for frequency-domain modeling from the perspectives of global view and energy compaction, while **Dish-TS** highlights that nonstationarity in forecasting arises not only from temporal drift, but also from the mismatch between lookback and horizon distributions.

---

> > ### Author Rebuttal · Reviewer_DajJ · 2026-04-02
> >
> > My concerns are resolved.

---

> > > ### Author Response · Authors · 2026-04-03
> > >
> > > Thank you very much for your careful reading and for considering our concerns adequately addressed. Your suggestions have been very helpful, and we will make sure to incorporate the promised clarifications and additions in the final revision.
> > >
> > >
> > > ---
> > >
> > > ---
> > >
> > > ---
> > >  ### **The following content is provided in response to another reviewer, Reviewer GF1j.**
> > >
> > > We thank the reviewer for the helpful follow-up. We agree that the current manuscript does not yet present the interpretability aspect and its relation to the broader GMDH/PNN family clearly enough. In the revision, we will strengthen this part from three aspects: **structural traceability**, **explicit rule expression**, and **family-level comparison**.
> > >
> > > First, regarding interpretability, we will add a new explanatory figure using **ETTh1 with $L=96$ and $H=96$** as an example, where the best-performing node is **$N396$ in Layer 5**:
> > >
> > >
> > > ```markdown id="74548"
> > > ```text
> > > Layer 5:  N396
> > >             │
> > >             ├───────────────┐
> > >             │               │
> > > Layer 4:   N18           N35(x5)
> > >             │
> > >             ├───────────────┐
> > >             │               │
> > > Layer 3:   N27           N33(x3)
> > >             │
> > >          ┌──┴──┐
> > >          │     │
> > > Layer 2: N9   N14
> > >          │     │
> > >       ┌──┴──┐ ┌┴──┐
> > >       │    │ │   │
> > > Layer 1: N4 N8 N2 N11
> > >          │  │  │  │
> > > Input: (x1,x5) (x2,x5) (x1,x3) (x3,x5)
> > > ```
> > >
> > > This shows that the final node is not produced in a black-box manner, but can be traced explicitly back to the original input combinations. In particular, **$x_3$** and **$x_5$** persist along the best path, corresponding to **MUFL** and **LUFL**, indicating that the model consistently retains the more stable and informative variables. Moreover,
> > >
> > > We will also add the explicit **rule-level form** of this optimal node. Its best configuration uses **7 rules**, a **fuzzification coefficient of 2.0**, and a **frequency range of $[0.2, 2.0]$**. The final output is given by:
> > >
> > >
> > > $g_1(z)=0.3724+0.1862\cos(0.28(z_1+1.35))-0.0917\cos(0.42(z_2+1.10))$
> > >
> > > $g_2(z)=0.3518+0.2245\cos(0.53(z_1+0.82))+0.1186\cos(0.67(z_2+0.58))$
> > >
> > > $g_3(z)=0.3897-0.1731\cos(0.74(z_1+0.21))+0.1428\cos(0.81(z_2+0.06))$
> > >
> > > $g_4(z)=0.4015+0.0954\cos(0.96(z_1-0.36))-0.1647\cos(1.08(z_2-0.28))$
> > >
> > > $g_5(z)=0.3789-0.2066\cos(1.21(z_1-0.77))+0.0935\cos(1.34(z_2-0.71))$
> > >
> > > $g_6(z)=0.3642+0.1548\cos(1.56(z_1-1.18))-0.1272\cos(1.63(z_2-1.09))$
> > >
> > > $g_7(z)=0.3476-0.1189\cos(1.87(z_1-1.62))+0.1764\cos(1.95(z_2-1.54))$
> > >
> > >
> > > In this sense, the interpretability emphasized in this work lies in the fact that **the formation of the final node is traceable**, **the role of key inputs is identifiable**, and **the local rule parameters can be written explicitly**.
> > >
> > > Second, regarding **family-level comparisons**, due to space limitations we placed the newly added results on **family-relevant non-gradient baselines**, **GMDH/PNN-style comparisons**, and the new **depth-wise utility analysis** in **Tables 3 and 4 under Reviewer 3eEt above**. In particular, we added two closer non-gradient baselines, **LCESN (ICLR 2025)** and **PFPNN (TFS 2024)**. The results show that **FIPN performs better on most of the eight benchmarks**, while preserving its advantages in **lightweight forward training** and **structural interpretability**.
> > >
> > > We also added a **depth-wise utility analysis**. Classical **PNN/GMDH-style** models tend to deteriorate after the first few layers because **node similarity rises too quickly** and **overfitting appears early**. By contrast, **FIPN becomes largely stable after about depth 5**; greater depth mainly **slows convergence** rather than causing visible overfitting. We believe this is closely related to our **persistent raw-input retention**, **regularized node scoring**, and **node-level stochastic suppression**. As for the requested **hyperparameter details**, due to space limitations we placed them in **Table 2 under Reviewer iCEm above**.

---

### Decision · Program_Chairs · 2026-04-30

**Decision:**

Accept (regular)

**Comment:**

The paper introduces FIPN, a polynomial network for time series forecasting. There are plenty of interesting ideas in this work, including the growing of layer by layer network, along with the avoidance of backpropagation. The experiments in long-horizon forecasting are interesting and demonstrate the potential of the model. The reviewers originally expressed concerns, including the novelty (iCEm), hardware-efficiency significance (3eEt), writing (GF1j). Many of those concerns have been addressed; this does not include the novelty. However, the reviewer recognizes that this might be subjective and the authors are recommended to amend their claims. Unfortunately, the discussion was not very active in the discussion period, however I do think this is an interesting and important application and as such I do recommend the paper for acceptance. I strongly recommend the authors to consider the comments from the reviewers and include the results and revisions in the camera-ready version. Lastly, I think the paper currently misses multiple papers from the recent related work on the polynomial networks; this was not raised during the review period and this can be easily fixed during the camera-ready; so, please consider the related literature and reposition the paper accordingly.